# QUANTUM-PEFT: ULTRA PARAMETER-EFFICIENT FINE-TUNING

**Toshiaki Koike-Akino**[1]* **Francesco Tonin**[2]* **Yongtao Wu**[2]
**Frank Zhengqing Wu**[2] **Leyla Naz Candogan**[2] **Volkan Cevher**[2]

[1] Mitsubishi Electric Research Laboratories (MERL), 201 Broadway, Cambridge, MA, USA
[2] LIONS, École Polytechnique Fédérale de Lausanne (EPFL), Lausanne, Switzerland

## ABSTRACT

This paper introduces Quantum-PEFT that leverages quantum computations for parameter-efficient fine-tuning (PEFT). Unlike other additive PEFT methods, such as low-rank adaptation (LoRA), Quantum-PEFT exploits an underlying full-rank yet surprisingly parameter-efficient *quantum unitary parameterization*. With the use of Pauli parameterization, the number of trainable parameters grows only logarithmically with the ambient dimension, as opposed to linearly as in LoRA-based PEFT methods. Quantum-PEFT achieves vanishingly smaller number of trainable parameters than the lowest-rank LoRA as dimensions grow, enhancing parameter efficiency while maintaining a competitive performance. We apply Quantum-PEFT to several transfer learning benchmarks in language and vision, demonstrating significant advantages in parameter efficiency.

## 1 INTRODUCTION

Fine-tuning large pre-trained models is a cost-effective method to adapt a general-purpose model to additional domains and tasks in computer vision and natural language processing (Devlin et al., 2018; Liu et al., 2019; He et al., 2021b; Radford et al., 2019; Brown et al., 2020; Dubey et al., 2024). Yet, even the practice of fine-tuning for each application can be costly as models scale to billions or trillions of parameters. The substantial memory requirements, such as GPT-3's 350GB footprint (Brown et al., 2020), can pose significant resource challenges, restricting practical deployment.

Parameter-efficient fine-tuning (PEFT) addresses the resource challenges of task specialization for massive pre-trained networks without the need to fine-tune parameters in full model weights dimensions (Aghajanyan et al., 2021; Hu et al., 2021; Edalati et al., 2023). For instance, low-rank adaptation (LoRA) (Hu et al., 2021) uses low-rank decompositions to modify weights, whereby reducing the number of trainable parameters. Despite its efficiency, there are limitations to the number of parameters, which include a compression ratio constrained by rank-1 decompositions and a linear scaling of trainable parameters with weight matrix dimensions.

We introduce Quantum-PEFT, a novel framework that achieves extremely parameter-efficient fine-tuning beyond LoRA-variants, e.g., (Zhang et al., 2023; Qiu et al., 2023; Liu et al., 2023b; Yeh et al., 2024), by leveraging quantum unitary parameterizations (Biamonte et al., 2017; Schuld et al., 2015). Quantum circuits are a way to represent unitary matrices as a product of smaller unitary matrices (i.e., quantum gates) with a total number of parameters growing logarithmically with the dimension, offering an extremely efficient framework for effective PEFT parameterizations. The core idea is to reparameterize the layers of pre-trained networks as generalized quantum circuits capturing complex transformations, which then only require a logarithmic number of trainable parameters. The ultra parameter efficiency is enabled by parameterizing the low-rank subspaces via Kronecker products of generalized Pauli rotations. The key contributions of our work include:

- We introduce new quantum-inspired modules based on generalized Pauli parametrization and quantum tensor network.

---

*Equal contribution. Correspondence to: Toshiaki Koike-Akino, Francesco Tonin <koike@merl.com, francesco.tonin@epfl.ch>.

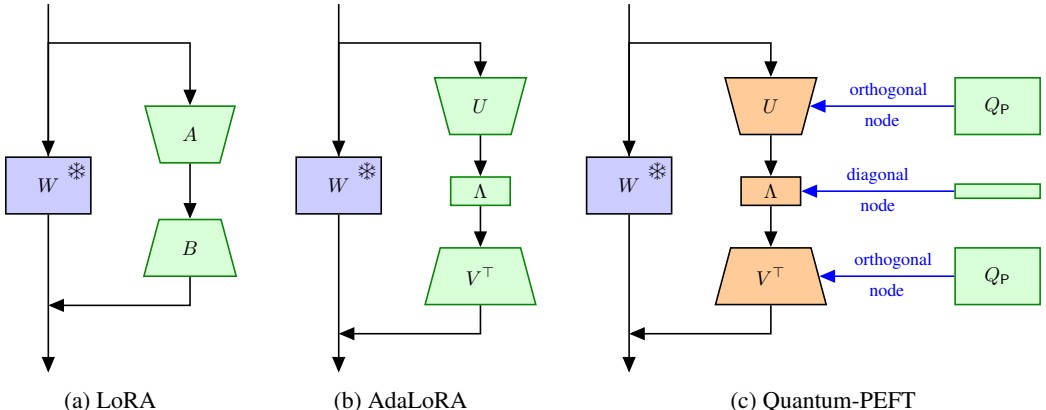

(a) LoRA        (b) AdaLoRA        (c) Quantum-PEFT

Figure 1: Overview of Quantum-PEFT compared to LoRA and AdaLoRA for PEFT. $W$ is the frozen pretrained weight, green boxes represent trainable parameters. LoRA updates $W$ by training the low-rank matrices $A, B$. AdaLoRA introduces the SVD trainable form $U, \Lambda, V$ with regularizer $\left\|U^\top U - I\right\|^2 + \left\|V^\top V - I\right\|^2$. In Quantum-PEFT, the matrices $U, V$ are not trainable parameters, but rather computed through quantum mappings of *orders-of-magnitude smaller* intrinsic parameters. Contrary to AdaLoRA, $U, V$ are left-orthogonal by construction in Quantum-PEFT.

- We propose a novel framework, named Quantum-PEFT, that leverages quantum unitary parameterizations for extremely parameter-efficient fine-tuning, achieving orders-of-magnitudes higher compression rates over state-of-the-art PEFT methods.

- Quantum-PEFT with Pauli parameterization enables logarithmic scaling of trainable parameters with respect to the ambient dimension of the model, realizing even smaller number of parameters than the lowest-rank LoRA.

- Through extensive experiments on language and vision tasks, we show Quantum-PEFT's significant advantage in parameter efficiency, achieving 5 to 25-fold reduction in trainable parameters compared to LoRA, yet maintaining competitive performance.

## 2 RELATED WORK

**Parameter-efficient fine-tuning (PEFT)** Parameter-efficient fine-tuning (PEFT) methods allow significantly lower model training cost for different downstream tasks. A plethora of methods have been proposed for PEFT (Houlsby et al., 2019; Aghajanyan et al., 2021; Hu et al., 2021; Edalati et al., 2023; Lester et al., 2021; Li and Liang, 2021; He et al., 2021a; Karimi Mahabadi et al., 2021; Chen et al., 2022; Jie and Deng, 2023; Hao et al., 2022; Houlsby et al., 2019; Pfeiffer et al., 2021), among which reparameterization-based techniques (Aghajanyan et al., 2021; Hu et al., 2021; Edalati et al., 2023) bear the most relevance to our study, where the model architecture is not changed but reparametrized with a lower number of trainable parameters. LoRA (Hu et al., 2021) updates the pretrained weight matrix through the addition of a product of two low-rank matrices with widespread adoption (Zi et al., 2023; Chavan et al., 2023; Hayou et al., 2024; Zhu et al., 2024). Many variants were introduced, e.g., tensor factorization (Edalati et al., 2023; Bershatsky et al., 2024; Chen et al., 2024a), nonlinear mappings (Liu et al., 2023a), Hadamard (Yeh et al., 2024) and Kronecker product (Edalati et al., 2023), layer sampling (Pan et al., 2024), embedding finetuning (Wu et al., 2024), and high-rank updates (Jiang et al., 2024b; Chen et al., 2024b). Additional discussions in Appendix A.6.

**Unitary-constrained PEFT** AdaLoRA (Zhang et al., 2023) introduces dynamic rank adjustment during fine-tuning, with additional regularizer for orthogonality. Unlike AdaLoRA with inexact orthogonality and extra regularization, we directly parameterize full-rank unitary matrices via efficient quantum circuit embeddings. Orthogonal fine-tuning (OFT) (Qiu et al., 2023; Liu et al., 2023b) employs a unitary matrix to transform the pretrained weights. OFT typically requires more trainable parameters and rely on expensive Cayley transform, highlighting the need for more efficient methods.

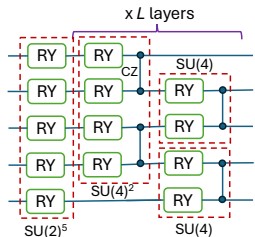

(a) Simplified two-design ansatz

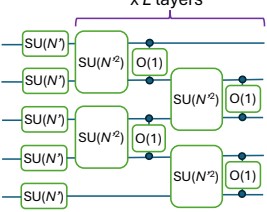

(b) Generalized Network

Figure 2: QML components. (a) Simplified two-design ansatz as (2). It alternates RY and CZ, i.e., a product of small unitary matrices. (b) Generalized quantum-inspired network for our unitary node. It generalizes the two-design ansatz to arbitrary dimensions by employing $\mathrm{SU}(N')$ blocks.

**Unitary-constrained machine learning** Unitary constraints can make training more stable and improve generalization, e.g., (Arjovsky et al., 2016; Jing et al., 2017; Chang and Wang, 2021). Different parametrizations are used, e.g., orthogonal weights through the Cayley transform (Helfrich et al., 2018) and Householder reflection (Mhammedi et al., 2017; Huang et al., 2022). Stiefel manifold optimization in ML is widely studied (Wisdom et al., 2016; Bansal et al., 2018; Li et al., 2019). Riemannian optimization has been considered for low-rank neural networks (Schotthöfer et al., 2022; Zangrando et al., 2024; Schotthöfer and Laiu, 2024). where they directly optimize the low-rank network factors with manifold optimization. Our approach differs as we do not optimize over the orthogonal factors directly, but rather parameterize them through trainable intrinsic weights.

**Quantum machine learning** Relevant quantum machine learning (QML) concepts include expressibility and entangling (Sim et al., 2019; Pérez-Salinas et al., 2020). Variational principles for quantum neural networks are in (Farhi and Neven, 2018), with extensions for quanvolutional networks (Henderson et al., 2020), quantum autoencoders (Romero et al., 2017), support vector machines (Suykens, 2013; Rebentrost et al., 2014), quantum graph neural networks (Zheng et al., 2021), and quantum generative adversarial networks (Lloyd and Weedbrook, 2018; Dallaire-Demers and Killoran, 2018). Quantum circuits are analytically differentiable enabling gradient optimization (Schuld et al., 2019).

## 3 PRELIMINARIES

**Notations:** Let $\mathrm{SU}(N)$, $\mathfrak{su}_N$, $\mathrm{SO}(N)$, $\mathrm{O}(N)$, and $\mathcal{V}_K(N)$ denote the special unitary Lie group of size $N$, its Lie algebra, special orthogonal group, orthogonal group, and real-valued Stiefel manifold having orthonormal $K$ frames $\mathcal{V}_K(N) = \{X \in \mathbb{R}^{N \times K} | X^\top X = I_K\}$, respectively. We denote $I$, $\mathbb{R}$, $\otimes$, $[\cdot]^\top$, and $\jmath$ as identity matrix of proper size, real numbers field, Kronecker product, transpose, and imaginary number, respectively. Let $L$, $q$, $N'$, $K'$ be the number of alternating entanglement layers, the number of qubits, orthogonal node size, and intrinsic rank, respectively. We denote the derived unitary matrices as $Q$, e.g, $Q_\mathsf{P}$ denotes the unitary Pauli-parameterized matrix. $\theta$ represents angles in Pauli parametrization, which are trainable parameters. $\mathsf{RY}(\theta)$ and $\mathsf{CZ}$ represent the quantum RY rotation gate and controlled-Z gate, respectively. Detailed list of symbols is in Appendix A.7.

In QML, neural network modules are realized by embedding classical data and weight values as quantum variational parameters such as Pauli rotation angles to control measurement outcomes. QML provides universal approximation property (Pérez-Salinas et al., 2020) and exponentially rich expressibity (Sim et al., 2019). Any quantum circuit can be decomposed (Kitaev, 1997) into a series of single-qubit rotations and two-qubit entanglements.

**Pauli matrices** Pauli operators play an important role to generate any unitary rotations up to a global phase. The group $\mathrm{SU}(N)$—the Lie group of unitary $N \times N$ matrices having determinant 1—can be generated by the Lie algebra $\mathfrak{su}_N$, i.e., the set of $N \times N$ skew-Hermitian matrices. For single-qubit rotations over $\mathrm{SU}(2)$, the Lie algebra is a span of $\{\jmath X, \jmath Y, \jmath Z\}$, with Pauli matrices: $X = \begin{bmatrix} 0 & 1 \\ 1 & 0 \end{bmatrix}, Y = \begin{bmatrix} 0 & -\jmath \\ \jmath & 0 \end{bmatrix}, Z = \begin{bmatrix} 1 & 0 \\ 0 & -1 \end{bmatrix}$. The exponential mapping of its linear combinations generates $\mathrm{SU}(2)$. For example, quantum RY rotation gate is given as

$$\mathsf{RY}(\theta) = \exp(-\jmath \tfrac{\theta}{2} Y) = \exp\left( \begin{bmatrix} 0 & -\theta/2 \\ \theta/2 & 0 \end{bmatrix} \right) = \begin{bmatrix} \cos(\theta/2) & -\sin(\theta/2) \\ \sin(\theta/2) & \cos(\theta/2) \end{bmatrix}, \qquad (1)$$

which alone spans the special orthogonal group $\mathrm{SO}(2)$ and forms $\mathrm{O}(2)$ along with a reflection $Z$.

**Quantum circuits and matrix operations**    A quantum circuit consists of a sequence of quantum gates applied to a quantum state. Mathematically, a quantum circuit can be viewed as a unitary matrix transforming one state (i.e., a tensor) to another tensor. The quantum circuit is a product of smaller unitary matrices (individual gates). We primarily employ two gates: single-qubit rotations RY gates, represented by $2 \times 2$ unitary matrices, and entangling 2-qubit CZ gates ($4 \times 4$ diagonal unitary matrices). A quantum circuit is built from a series of gates and can represent any unitary matrix.

**Two-design ansatz**    In QML, two-design ansatz (Cerezo et al., 2021) use a small number of parameters in order of $\mathcal{O}[\log_2(N)]$ to represent unitary matrices $\mathrm{SU}(N)$ whose statistical properties are identical to ensemble random unitaries. This property suggests that gradient optimization can uniformly adjust few-parameter Pauli rotation angles along the unitary group $\mathrm{SU}(N)$. Comparing to the full degree of freedoms of $\dim[\mathrm{SU}(N)] = N^2 - 1$ for any skew-Hermitian matrices, the QML has a great potential to realize parameter-efficient representation in its logarithmic order. In the following, we introduce a generalized framework to extend the QML features for extremely parameter-efficient neural network modules, which constitute the building blocks of Quantum-PEFT.

## 4    QUANTUM-PEFT

We propose to parameterize the matrix added to the pretrained weights $W$ by leveraging an ultra compact representation based on quantum unitary parameterizations. Similarly to AdaLoRA (Zhang et al., 2023), we reparametrize the weight updates as a product of unitary matrices $U \in \mathcal{V}_K(N) \subset \mathbb{R}^{N \times K}, V \in \mathcal{V}_K(M) \subset \mathbb{R}^{M \times K}$ and a diagonal matrix $\Lambda \in \mathbb{R}^{K \times K}$. Specifically, the weight update $\Delta W$ for a weight matrix $W \in \mathbb{R}^{N \times M}$ is given by: $\Delta W = U \Lambda V^{\top}$. In Quantum-PEFT, the low-rank $U, V$ are not optimization variables, but rather are expressed via efficient unitary parameterizations, i.e., Kronecker products of generalized Pauli rotations. In this way, the number of trainable parameters depends on the chosen underlying unitary parametrization. Our PEFT pipeline is shown in Figure 1.

### 4.1    QUANTUM-PEFT: PAULI, ORTHOGONAL AND DIAGONAL NODES

In this section, we introduce generalized quantum-inspired modules as the core building blocks of Quantum-PEFT, i.e., product of RY and CZ gates for trainable orthogonal nodes (akin to generalized RY modules), and trainable diagonal nodes (akin to generalized CZ modules). We additionally show how to solve the power-of-two scaling limitation of QML and address efficient computation.

**Pauli parameterization**    The simplified two-design ansatz (Cerezo et al., 2021) visualized in Figure 2a in Appendix uses an alternating circuit composed of RY and controlled-Z (CZ) entangling gates: $\mathsf{CZ} = \mathrm{diag}[1, 1, 1, -1]$, which is an element of reflection groups $\mathrm{O}(1)^4 = \{\pm 1\}^4$. This ansatz is suited for neural networks as they are real-valued quantum operations over $\mathrm{SO}(N)$, i.e., not complex-valued operations over $\mathrm{SU}(N)$ arising when using RZ or RX rotations. In Quantum-PEFT, we propose to use the Pauli parameterization based on this construction, as given below:

$$Q_{\mathsf{P}} = \prod_{l=1}^{L} \left( \left( I \otimes \left( \mathsf{CZ}^{\otimes \frac{q-1}{2}} \bigotimes_{k=2}^{q} \mathsf{RY}(\theta_{k,2l+1}) \right) \right) \left( \left( \mathsf{CZ}^{\otimes \frac{q-1}{2}} \bigotimes_{k=1}^{q-1} \mathsf{RY}(\theta_{k,2l}) \right) \otimes I \right) \right) \bigotimes_{k=1}^{q} \left( \mathsf{RY}(\theta_{k,1}) \right),$$
(2)

where $L$ is the number of alternating entanglement layers, and $q = \log_2(N)$ is the number of qubits; in (2) $q$ is odd s.t. $(q-1)/2$ is integer and the $q$ even case can be trated similarly. In (2), the trainable parameters are the rotation angles $\theta_{k,l}$ associated with the employed RY gates (1). The $\theta_{k,l}$ for each RY gates is trainable, where the RY gate (1) has a single parameter. This Pauli parameterization therefore has $(2L+1)\log_2(N) - 2L$ parameters, increasing only logarithmically with the matrix size $N$. The entangling capacity is controlled by $L$ and is further discussed in Appendix A.4. While the tensor rank is 2, thanks to the alternating CZ entanglement, the effective rank of the matrix $Q_{\mathsf{P}}$ is of full $N$ contrary to low-rank AdaLoRA (Zhang et al., 2023). Not only parameter efficient, but Pauli parametrization is also computationally efficient as it takes $\mathcal{O}[N \log_2(N)L]$ operations compared to quadratic complexity for unitary matrix rotations. Motivated by this quantum-inspired network, we further generalize the parameterization from $\mathrm{SU}(2)$ to $\mathrm{SU}(N')$ with an arbitrary size of $N' > 2$ as

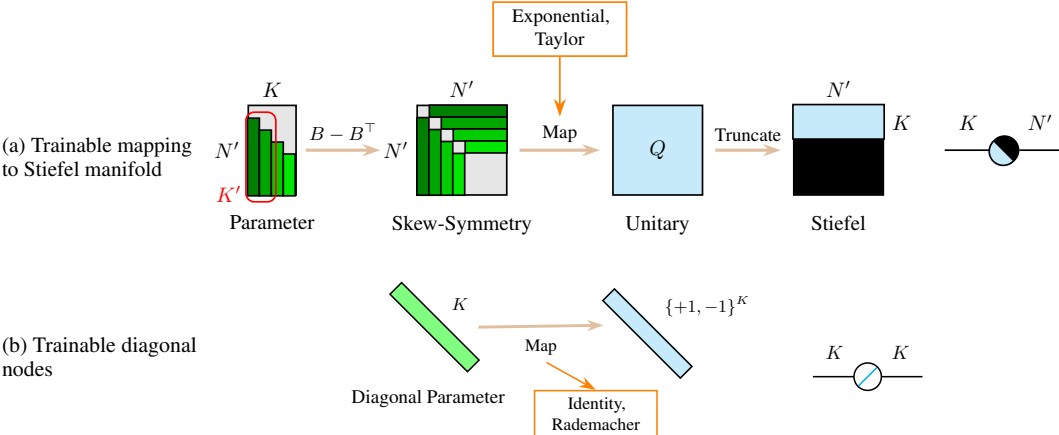

Figure 3: Quantum-PEFT modules with corresponding tensor diagrams. (a) Trainable mapping onto the Stiefel manifold $\mathcal{V}_K(N')$. Intrinsic rank $K'$: top $K'$ columns are trainable parameters in $B$. (b) Generalized CZ modules for diagonal nodes on either $\mathrm{O}(1)^{N'}$ or $\mathbb{R}^{N'}$.

shown in Figure 2b as a building block to *represent a large unitary matrix* $\mathrm{SU}(N)$ *with a smaller number of unitary factors* $\mathrm{SU}(N')$ *in a logarithmic scale of* $\mathcal{O}[\log_{N'}(N)]$. To this end, we show how to $(i)$ generalize RY gates by generating unitary matrices from skew-symmetric matrices, and $(ii)$ use recursive cosine-sine decomposition to allow non-power-of-two $N$.

**Mapping to Stiefel manifold**  Arbitrary unitary rotations of size $N'$ can be realized by mapping skew-Hermitian matrices for $\mathrm{SU}(N')$ or skew-symmetric matrices for $\mathrm{SO}(N')$. We consider mapping methods for orthogonal nodes below. Let $B \in \mathbb{R}^{N' \times N'}$ be a strictly lower-triangular matrix with non-zero trainable parameters from $B_K \in \mathbb{R}^{N' \times K}$ with $K \leq N'(N'-1)/2$. Given a skew-symmetric matrix $A = B - B^\top \in \mathbb{R}^{N' \times N'}$, we can generate a corresponding unitary (orthogonal) matrix, e.g., with exponential mapping or Taylor series

$$Q'_\mathsf{E} = \exp(A), \qquad Q'_\mathsf{T} = \sum_{p=0}^{P} \frac{1}{p!} A^p, \tag{3}$$

where the mapping $Q'_\mathsf{T}$ is an approximated version of $Q'_\mathsf{E}$ up to a polynomial order $P$ to avoid matrix exponentiation. Diverse unitary parameterizations are possible, e.g., Cayley transform, Householder reflection, Givens rotation; we focus on $Q'_\mathsf{T}$ as it shows a good trade-off between accuracy, speed, and parameter efficiency. We refer to Appendix A.1 for detailed comparisons and discussions.

Figure 3(a) illustrates the pipeline to generate matrices on the Stiefel manifold $\mathcal{V}_K(N')$ from trainable parameters through the exponential/Taylor mapping. After mapping skew-symmetric matrix, truncating the square unitary matrix as $Q_{:K,:}$ generates a right-orthogonal matrix. As all the mappings described above are differentiable, the Lie algebra can be trained via gradient methods, and we use PyTorch's autograd to compute the backward pass gradient. QML literature has employed the gradient method successfully (You et al., 2023; Bermejo et al., 2024) and we empirically observed no issues in loss convergence. While most mapping methods are studied in other literature (Qiu et al., 2023; Liu et al., 2023b; Chang and Wang, 2021; Wisdom et al., 2016; Bansal et al., 2018; Li et al., 2019), in a PEFT context we can further reduce the number of parameters by masking out the Lie parameters. For example, only the top $K'$ columns of $B_K$ are trainable, while the other parameters are frozen or null-out. We call $K'$ an intrinsic rank to cover a subset of $\mathcal{V}_K(N')$.

A naive implementation of the above mapping pipeline can use redundant memory before truncation. We resolve the memory redundancy by tensor contraction ordering (Pfeifer et al., 2014), except for $Q'_\mathsf{E}$, e.g., multiplying unitary matrix with a feature vector $x \in \mathbb{R}^{N'}$ is recursively contracted as $Q'_\mathsf{T} x = \sum_p \frac{1}{p!}(B - B^\top)^p x$, which does not require the full matrix $Q'_\mathsf{T}$ but only a series of low-rank multiplications with $B$. Computational complexity of the mapping and numerical accuracy considerations are discussed in the next subsection.

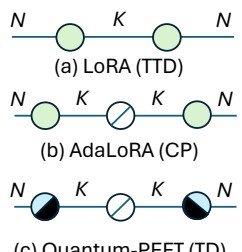

(a) LoRA (TTD)

(b) AdaLoRA (CP)

(c) Quantum-PEFT (TD)

Figure 4: Tensor diagram of LoRA variants.

Table 1: Memory requirements to store trained LoRA and $Q_P$ weights for DeBERTaV3 base, Llama 3.1 405B, and GPT-4.

| | Rank | LoRA | | Quantum-PEFT | |
|---|---|---|---|---|---|
| | | # Parameters | Required Bytes | # Parameters | Required Bytes |
| BASE | 1 | 36.9K | 0.14MB | 3.69K | 0.01MB |
| | 16 | 589.8K | 2.25MB | 3.98K | 0.02MB |
| | 256 | 9437.2K | 36.00MB | 9.7K | 0.04MB |
| LLAMA | 1 | 8.26M | 31.51MB | 60.7K | 0.23MB |
| | 16 | 132.1M | 0.50GB | 64.5K | 0.25MB |
| | 256 | 2188.2M | 8.35GB | 127.3K | 0.49MB |
| GPT-4 | 1 | 36.7M | 140MB | 269.7K | 1.03MB |
| | 16 | 586.6M | 2.24GB | 286.4K | 1.09MB |
| | 256 | 9385.6M | 35.80GB | 565.1K | 2.15MB |

**Overcoming the power-of-$N'$ limitation** The pipeline in Figure 3(a) can be seen as generalized RY modules for $\mathrm{SU}(N')$, then assembled to construct a larger unitary node $\mathrm{SU}(N)$ as visualized in Figure 2b, meant to be used within $Q_P$ s.t. Quantum-PEFT can handle arbitrary dimensions $N' > 2$. However, $N$ should be a power of $N'$. Using quantum Shannon decomposition (QSD) (Shende et al., 2005) i.e. recursive cosine-sine decomposition (CSD), any unitary matrix $\mathrm{SU}(N)$ can be constructed by $\mathrm{SU}(N_1)$ and $\mathrm{SU}(N_2)$ for lower dimensions such that $N_1 \geq N_2$ and $N_1 + N_2 = N$ for $N > 1$:

$$U = \begin{bmatrix} U_1 & 0 \\ 0 & U_2 \end{bmatrix} \begin{bmatrix} C & -S & 0 \\ 0 & 0 & I \\ S & C & 0 \end{bmatrix} \begin{bmatrix} V_1 & 0 \\ 0 & V_2 \end{bmatrix}, \tag{4}$$

where $U \in \mathrm{SU}(N)$, $U_1, V_2 \in \mathrm{SU}(N_1)$, $U_2, V_1 \in \mathrm{SU}(N_2)$, diagonal cosine and sine matrices such that $C^2 + S^2 = I \in \mathbb{R}^{N_2 \times N_2}$. Hence, power-of-$N'$ rotations such as Kronecker products of Pauli rotations can be still used for arbitrary size of matrices. It hence can solve the power-of-$N'$ limitation.

*Example* 4.1. In the simple case $N' = 2$, $N_1$ and $N_2$ are adjustable parameters s.t. $N = N_1 + N_2$ with non-power-of-two $N$. For example, when $N = 12$, we can use $N_1 = 8 = 2^3$ and $N_2 = 4 = 2^2$, where we can use four power-of-two unitary matrices of $U_1, U_2, V_1, V_2$ as well as cos-sine RY rotations. QSD allows recursive decomposition. For example, when $N = 28$, we apply cos-sin decomposition twice to have $N = 2^4 + 2^3 + 2^2$, where the first $(N_1, N_2) = (2^4, 2^3 + 2^2)$ and the second CSD for the $N_2$ part will be further decomposed as $(N_1, N_2) = (2^3, 2^2)$.

**Diagonal node** Generalizing CZ modules provides a few options: trainable diagonal matrix in any real number $\mathbb{R}^K$, discrete number, and binary $\{\pm 1\}^K$. Trainable discrete diagonal matrix can be realized e.g. by Gumbel softmax or ReinMax trick (Liu et al., 2024). We refer to a trainable binary diagonal matrix as Rademacher mapping, which can create perfect unitarity and reflection group in $\mathrm{O}(1)^K$. Specifically, Rademacher mapping with ReinMax trick is given as $Q_R = \mathrm{diag}[\mathrm{ReinMax}_\tau([\Lambda, -\Lambda]) \times [+1, -1]]$ with a temperature $\tau$ and diagonal parameter $\Lambda \in \mathbb{R}^K$. Fig. 3(b) illustrates diagonal nodes and its tensor diagram. When identity map is used, it can be used as singular values of any matrices under its singular-value decomposition (SVD). Therefore, the use of both trainable unitary matrices and diagonal matrices is sufficient for general representation of any matrix through its SVD , solving the only-unitarity limitation of typical QML.

## 4.2 QUANTUM-PEFT: ANALYSIS

Quantum-PEFT leverages a parameter-efficient network by exploiting the new modules: trainable small orthogonal matrices parameterized by the Lie algebra to generate Stiefel manifold $\mathcal{V}_K(N)$ via our generalized RY modules; trainable diagonal nodes either on $\mathbb{R}^K$ or $\mathrm{O}(1)^K$ via our generalized CZ modules; and the Pauli parameterization. We now analyze the orders-of-magnitude improvements in parameter efficiency w.r.t. existing LoRA variants and give more discussions.

**Parameter efficiency** LoRA uses two $K$-rank matrices, having $2NK$ parameters. This is known as 2-mode tensor train decomposition (TTD). Any TTD can be transformed into its canonical form, having multiple unitary nodes and one diagonal node. AdaLoRA uses approximated SVD, where unitarity

is not perfectly imposed, leading to $K(K+1)$ redundant parameters and extra regularization terms. From the tensor network perspective, AdaLoRA falls under Canonical Polyadic (CP) decomposition which does not strictly assume orthogonality. Using the Lie algebra, Quantum-PEFT can readily realize the non-redundant parameterization for trainable SVD (i.e., 2-mode Tucker decomposition: TD). In this form, the number of trainable parameters depends on the chosen underlying parametrization for the orthogonal matrices. Specifically, the Taylor parametrization $Q'_\mathsf{T}$ for the maximum decomposition size $N' = N, K' = K$ yields $2NK - K^2$ trainable parameters, and the Pauli parametrization $Q_\mathsf{P}$ achieves an extremely compact representation with only $\mathcal{O}(2((2L + 1)\log_2(N) - 2L) + K)$ parameters , scaling logarithmically with the matrix dimension $N$. The underlying parameterizations induced by our generalized RY modules spanning orthogonal group can effectively capture a full-rank weight update. This contrasts with AdaLoRA (Zhang et al., 2023), which uses approximate orthogonality imposed by regularization terms in optimization, failing to reduce the number of trainable parameters being limited by the low-rank decomposition. Consequently, Quantum-PEFT enables orders-of-magnitude parameter reduction compared to conventional LoRA-based approaches. The theoretical memory requirements of PEFT applied to query/value weights in comparison to LoRA are shown in Table 1 and Figure 4 shows tensor diagrams under tensor network interpretation. More discussions of other tensor networks are found in Appendix A.3.

**Computational complexity**  Regarding computational complexity of computing the mapping, Pauli parameterization $Q_\mathsf{P}$ has runtime $\mathcal{O}((L + 1)N\log_2(N))$ with the efficient Kronecker Shuffle algorithm (Plateau, 1985), while the number of trainable parameters scales logarithmically with $N$. This shows that, up to small constant factors (e.g., $L$ can be set to 1 for PEFT with sensitivity analysis in Appendix A.4), the computational complexity remains highly competitive with LoRA's $\mathcal{O}(2NK)$ even with the mapping. Contrary to LoRA, the logarithmic scaling of parameters allowed by our construction translates to a substantial reduction in memory footprint, which becomes increasingly important when dealing with very large models. The Taylor mapping $Q'_\mathsf{T}$ has complexity $\mathcal{O}(2(P + 1)NK)$, yielding a comparable time complexity order with LoRA $\mathcal{O}(2NK)$ at $N = N', K = K'$. We remark that the Taylor parameterization can be used independently, namely $Q_\mathsf{T}$, to generate orthogonal matrices from underlying small trainable weights as in Figure 3(a), which is computationally cheap and results in $\mathcal{O}(2NK - K^2)$ parameters. This setup is preferred for reduced training time when more memory bandwidth is available, while $Q_\mathsf{P}$ provides the highest parameter savings.

**Quantization**  To further save memory, we can use a standard integer quantization for trainable parameter: $\theta$: $\theta_\mathsf{q} = \mathrm{round}((\theta - \mu)/\beta)\beta + \mu$, where scale value $\beta = (\theta_{\max} - \theta_{\min})/(2^n - 1)$ and zero value $\mu = \theta_{\min}$ for $n$-bit quantization. The maximum $\theta_{\max}$ and minimum values $\theta_{\min}$ are obtained in a chunk of group size $g$. When the quantization is applied on the Lie parameters, we employ the straight-through trick for quantization-aware training (QAT), i.e., $\theta := \theta_\mathsf{q} + \theta - \theta.\mathrm{detach}()$, where $.\mathrm{detach}()$ means no gradient passing. Once trained, the required memory will be $n + 32/g$ bits per Lie parameter when $\beta$ and $\mu$ use floating-point (FP) 16 bits precision.

## 5 EXPERIMENTS

We evaluate our Quantum-PEFT for DeBERTaV3 (He et al., 2023), GPT-2 (Radford et al., 2019), ViT (Dosovitskiy et al., 2020), and Mistral-7B (Zhang et al., 2023) on diverse fine-tuning. We fine-tune (1) DeBERTaV3 and Mistral-7B on the General Language Understanding Evaluation (GLUE) benchmark (Wang et al., 2019); (2) GPT-2 Medium on E2E Challenge following the original LoRA paper (Hu et al., 2021); and (3) ViT on CIFAR10 (Krizhevsky et al., 2009). Our experiments are not to claim that Quantum-PEFT always improves accuracy w.r.t. LoRA, but to show that Quantum-PEFT can maintain a competitive level of accuracy with orders-of-magnitude fewer parameters.

### 5.1 GLUE BENCHMARK

The experiment is conducted on the GLUE benchmark, which consists of NLP understanding tasks. Our experiment follows the set-up in (Zhang et al., 2023). The fine-tuning is applied on DeBERTaV3-base (He et al., 2023). We compare Quantum-PEFT with the following baselines: Full parameters fine-tuning (FT), LoRA (Hu et al., 2021), BitFit (Zaken et al., 2022), adapter tuning with Houlsby adapter (HAdapter) (Houlsby et al., 2019), adapter tuning with Pfeiffer adapter (PAdapter) (Pfeiffer et al., 2021), AdaLoRA (Zhang et al., 2023), LoKr (Yeh et al., 2024), LoHa (Hao et al., 2022), MORA

Table 2: Results with DeBERTaV3 base on GLUE benchmark. We present the Matthew's correlation for CoLA, the average correlation for STS-B, and the accuracy for other tasks. In each column, the best-performing PEFT approach is highlighted in **bold** and the second best is underlined.

| Method | # Trainable Parameters | SST-2 | CoLA | RTE | MRPC | STS-B | Avg. | Memory |
|---|---|---|---|---|---|---|---|---|
| FT | 184M | 95.63 | 69.19 | 83.75 | 89.46 | 91.60 | 85.93 | 14200× |
| BitFit | 0.1M | 94.84 | 66.96 | 78.70 | 87.75 | 91.35 | 83.92 | 7.69× |
| HAdapter | 0.61M | 95.30 | 67.87 | 85.56 | 89.22 | 91.30 | 85.85 | 46.92× |
| PAdapter | 0.60M | 95.53 | 69.48 | 84.12 | 89.22 | 91.52 | 85.97 | 46.15× |
| HAdapter | 0.31M | 95.41 | 67.65 | 83.39 | 89.25 | 91.31 | 85.40 | 23.85× |
| PAdapter | 0.30M | 94.72 | 69.06 | 84.48 | 89.71 | 91.38 | 85.87 | 23.08× |
| LoRA | 0.33M | 94.95 | 68.71 | 85.56 | 89.71 | **91.68** | 86.12 | 25.38× |
| AdaLoRA | 0.32M | 95.80 | **70.04** | **87.36** | 90.44 | 91.63 | **87.05** | 24.62× |
| LoHa | 0.33M | 95.50 | 66.52 | 80.43 | 89.95 | 89.46 | 84.37 | 25.38× |
| LoKr | 0.073M | 95.07 | 69.46 | 85.20 | 89.71 | 90.76 | 86.04 | 5.62× |
| MORA | 0.49M | 95.79 | 67.13 | 85.19 | 89.08 | 90.13 | 85.46 | 37.87x |
| QuanTA | 0.093M | 95.30 | 67.75 | 84.48 | 89.22 | 91.01 | 85.55 | 7.15x |
| Quantum-PEFT | **0.013M** | **95.85** | 67.85 | 86.57 | **90.78** | 91.06 | 86.42 | **1×** |

Table 3: Results for different adaptation methods on the E2E benchmark and GPT2 Medium model. Quantum-PEFT achieves similar performance as LoRA with 4 times less trainable parameters and better performance than LoKr with same parameters.

| Method | # Trainable Parameters | BLEU | NIST | METEOR | ROUGE-L | CIDEr |
|---|---|---|---|---|---|---|
| FT | 354.92M | 68.2 | 8.62 | 46.2 | 71.0 | 2.47 |
| LoRA | 0.39M | 66.88 | 8.55 | **45.48** | **68.40** | **2.31** |
| AdaLoRA | 0.38M | 64.64 | 8.38 | 43.49 | 65.90 | 2.18 |
| LoHa | 0.39M | 65.03 | 8.45 | 43.76 | 66.54 | 2.22 |
| LoKr | **0.098M** | 63.90 | 8.27 | 42.35 | 65.22 | 2.04 |
| Quantum-PEFT | **0.098M** | **67.46** | **8.58** | 45.02 | 67.36 | **2.31** |

(Jiang et al., 2024b), and QuanTA (Chen et al., 2024b). We fine-tune the query/key/value projection matrices, the output projection in the attention block, and the weight matrices in two-layer MLPs. For all of the baselines, we follow the hyperparameters in (Zhang et al., 2023). For Quantum-PEFT, we use $Q_{\mathsf{P}}$ with $L = 1$ in all tasks. We select the best learning rate by parameters sweep. We conduct five runs with different random seeds and report the mean. We use the same number of training epochs as in AdaLoRA. Due to limited computing resources, we focus on tasks with training instances less than 100k, including SST-2, CoLA, RTE, MRPC, and STS-B. Detailed setups are given in Appendix B.

The results are summarized in Table 2. We can see that in both SST-2 and MRPC tasks, Quantum-PEFT can outperform AdaLoRA. On other tasks, Quantum-PEFT can still achieve comparable performance with other baselines. Notably, Quantum-PEFT only requires 0.013 million parameters, which are 25 times fewer than LoRA.

## 5.2 E2E BENCHMARK

We fine-tune GPT-2 (Radford et al., 2019) Medium on the common E2E natural language generation benchmark (Novikova et al., 2017), following the setups of (Hu et al., 2021). GPT2-Medium has 354M parameters with 24 transformer layers. The E2E benchmark consists of 42,200 samples for training, 4,600 for validation, and 4,600 for testing. We compare LoRA (Hu et al., 2021),

Table 4: Efficiency comparison on GPT2-Medium.

| Resource | LoRA | AdaLoRA | LoHa | LoKr | Quantum-PEFT |
|---|---|---|---|---|---|
| Training Time (ms/batch) | 1719.06 | 1795.91 | 1874.04 | 1790.24 | 1723.39 |
| Memory Ratio | $4.03\times$ | $4.03\times$ | $4.03\times$ | $1\times$ | $1\times$ |

Table 5: Results with Mistral-7B on GLUE Benchmark. The reported metrics are as in Table 2.

| Method | # Trainable Parameters | SST-2 | CoLA | RTE | MRPC | STS-B | Avg. |
|---|---|---|---|---|---|---|---|
| LoRA | 3.54M | 96.21 | 68.83 | 87.46 | 87.74 | 90.70 | 86.19 |
| AdaLoRA | 3.54M | **96.90** | 70.38 | 87.53 | 89.52 | 90.96 | 87.06 |
| Quantum-PEFT | **0.758M** | 96.67 | **70.61** | **88.10** | **89.94** | **91.63** | **87.39** |

AdaLoRA (Zhang et al., 2023), LoKr (Yeh et al., 2024), LoHa (Hyeon-Woo et al., 2022), and full FT with Quantum-PEFTwith the simple independent Taylor parameterization $Q_T$, $P = 3$ for efficient computations at larger model size. Full FT results are sourced from prior works (Zi et al., 2023). For fair comparison, we use the same training settings and hardware, i.e., 4 NVIDIA A100 GPUs, for all methods. We train the baselines using the code provided by the respective authors or using the `peft` library from Hugging Face. We apply PEFT to the query and value projection layers in each attention block and use the same number of training epochs, batch size, and LoRA scaling, except different learning rate. Table with all hyperparameters is provided in Appendix B.

Table 3 shows the results for E2E Challenge dataset. Quantum-PEFT's performance is on par or better than LoRA with approximately 4 times less trainable parameters, and significantly beats LoKr with the same number of parameters. For the BLEU metric, our method obtains $0.58$ gain compared with LoRA, with comparable results on the other metrics. We report results from the final epoch, whereas Hu et al. (2021) presented the best performance observed during training, and used 4 GPUs rather than 1 due to time constraints, which may contribute to the observed variances w.r.t. the reported performance in (Hu et al., 2021). These results demonstrate that Quantum-PEFT can achieve a comparable level of accuracy to the baselines while using significantly fewer parameters. In Table 4, we evaluate the training time and memory benefits of our method over LoRA while fine-tuning GPT2-Medium. We find that using Quantum-PEFT results in similar training time to LoRA. W.r.t. memory requirements, we observe a 4x reduction in storage with Quantum-PEFT on GPT2-Medium.

## 5.3 LARGE-SCALE FINE-TUNING

To assess the effectiveness of Quantum-PEFT at larger model scales, we fine-tune the Mistral-7B model (Jiang et al., 2024a). Mistral-7B is a recent language model exhibiting strong performance at its size, outperforming the larger Llama 2-13B on many benchmarks (Jiang et al., 2024a; Zhang and Pilanci, 2024). We experiment our method with this new language model on the GLUE benchmark for natural language understanding problems. We use the LoRA-related hyperparameters as with the DeBERTaV3 experiments. We use the optimization setups from (Zhang and Pilanci, 2024), where for all methods we use 4-bit quantization, employ AdamW optimizer over 5 epochs, and also fine-tune the gate projection matrices. Table 5 presents the results. Quantum-PEFT significantly outperforms the strongest methods on GLUE from Section 5.1 on almost all datasets, while using $4.67\times$ fewer trainable parameters than LoRA. This further highlights Quantum-PEFT's parameter efficiency and even superior-than-LoRA performance when scaling to billion-scale large language models.

## 5.4 IMAGE CLASSIFICATION BENCHMARK

We evaluate a transfer learning task of the ViT model (`google/vit-base-patch16-224`) pre-trained on ImageNet-21k (Deng et al., 2009) towards CIFAR10 dataset (Krizhevsky et al., 2009). Detailed settings are found in Appendix B. The base model is frozen after being quantized with 3 bits, and adapters for query and value projections are updated. For Quantum-PEFT, we use $Q_P$ parameterization for $K = L = 1$. Table 6 shows the comparison of full FT, LoRA, and Quantum-PEFT. When no fine-tuning was applied, the classification accuracy of the original ViT is poor, and thus fine-tuning is important. Compared to the full FT which requires $95.81M$ parameters, PEFT can significantly reduce the required number of trainable parameters, especially with our Quantum-

Table 6: Results for ViT transfer learning from ImageNet-21k to CIFAR10. Base ViT 3-bit quantized.

| Method | Original | FT | LoRA$_{K=1}$ | LoRA$_{K=2}$ | LoRA$_{K=4}$ | Quantum-PEFT |
|---|---|---|---|---|---|---|
| # Parameters | — | 85.81M | 0.037M | 0.074M | 0.147M | **0.007M** |
| Accuracy | 76.21% | 98.05% | 98.14% | 98.30% | 98.39% | **98.46%** |

Table 7: Quantization impact on Lie parameters with Taylor parameterization for ViT transfer learning from ImageNet-21k to CIFAR10. Base ViT is not quantized.

| Quantization | FP32 | INT8 | INT4 | INT3 | INT2 | INT1 |
|---|---|---|---|---|---|---|
| # Bits per parameter | 32 | 8.25 | 4.25 | 3.25 | 2.25 | 1.25 |
| Accuracy (Uniform Bit Loading) | 98.81% | **98.79%** | 98.78% | 98.75% | 98.67% | 97.96% |
| Accuracy (Adaptive Bit Loading) | 98.81% | 98.78% | **98.87%** | **98.80%** | **98.77%** | **98.64%** |

PEFT. For example, Quantum-PEFT has 21-hold fewer parameters than LoRA with rank 4. More importantly, Quantum-PEFT shows superior performance despite the fact of the fewest parameters.

**Quantization** Table 7 shows the QAT performance with different number of bits per the Lie parameter for Taylor parameterization ($Q_\mathsf{T}$, $K = K' = 4$ and $P = 18$). Here, the base ViT model is not quantized, while only adapters are quantized. We use $g = 128$ and FP16 for scale and zero values $\beta$ and $\mu$. We observe that reducing the precision for the Lie parameterization can gradually degrade. Nevertheless, thanks to QAT, no significant loss can be seen even with 1-bit integer quantization from FP32: i.e., $0.65\%$ degradation. We also evaluate the performance of mixed-precision Taylor parameterization. One can see that adaptive bit loading can significantly improve the performance at few-bit quantization regimes. For instance, adaptive 1-bit quantization of Lie parameters has just $0.17\%$ loss from FP32 and $0.28\%$ improvement from uniform 1-bit quantization. This may come from the effective pruning gain. More details of quantization are given in Appendix A.3 and A.5.

**Sensitivity analysis of intrinsic rank** Our introduced intrinsic rank $K'$ can reduce the trainable parameters than the specified rank $K$, by masking the top $K'$ columns of Lie parameters. In Table 8, we show the impact of varying $K'$ in the same settings as Table 7 on the ViT transfer learning task. Decreasing $K'$ gradually reduces the required number of parameters. While the subspace rank is $K = 8$, the number of parameters can be effectively $K' \le K$. The performance degradation from $K' = 8$ to $K' = 1$ is only $0.49\%$, and more importantly the accuracy is still better than LoRA in Table 6. For example, LoRA with $K = 1$ has an accuracy of $98.14\%$, while Quantum-PEFT $Q_\mathsf{T}$ parameterization with $K = 8$ and $K' = 1$ has $98.38\%$, at the comparable number of parameters. It shows the great potential of masking out the Lie parameters while keeping higher subspace rank.

Table 8: Impact of intrinsic rank $K'$ for ViT transfer learning from ImageNet-21k to CIFAR10.

| Intrinsic rank $K'$ | 1 | 2 | 3 | 4 | 5 | 6 | 7 | 8 |
|---|---|---|---|---|---|---|---|---|
| # Parameters | 0.037M | 0.074M | 0.111M | 0.147M | 0.184M | 0.221M | 0.257M | 0.294M |
| Accuracy | 98.38% | 98.52% | 98.76% | 98.74% | 98.63% | 98.79% | 98.81% | 98.87% |

# 6 CONCLUSIONS

In this work, we introduced Quantum-PEFT, a novel framework leveraging quantum machine learning principles to achieve extremely parameter-efficient fine-tuning of large pre-trained models. Through reparameterization as generalized quantum circuits, Quantum-PEFT represents weight updates using highly compact unitary matrix embeddings. Quantum-PEFT can achieve even lower parameter number than the lowest-rank LoRA; unlike prior low-rank adaptation methods bottlenecked by linear parameter growth, the employed Pauli parametrization scales logarithmically with the model size. By use of QSD, our unitary node can use non-power-of-two dimensions. Our experiments across language and vision benchmarks validate Quantum-PEFT's excellent capabilities, achieving orders-of-magnitudes higher compression rates than LoRA while maintaining competitive performance.

ACKNOWLEDGEMENTS

LIONS-EPFL was supported by Hasler Foundation Program: Hasler Responsible AI (project number 21043). LIONS-EPFL was sponsored by the Army Research Office under Grant Number W911NF-24-1-0048. LIONS-EPFL was supported by the Swiss National Science Foundation (SNSF) under grant number 200021_205011.

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

## A  FURTHER DETAILS AND DISCUSSIONS ON QUANTUM-PEFT

To further elaborate on Quantum-PEFT, we provide the tensor network diagrams in Figure 5 exemplifying its mechanism w.r.t. other LoRA-based methods.

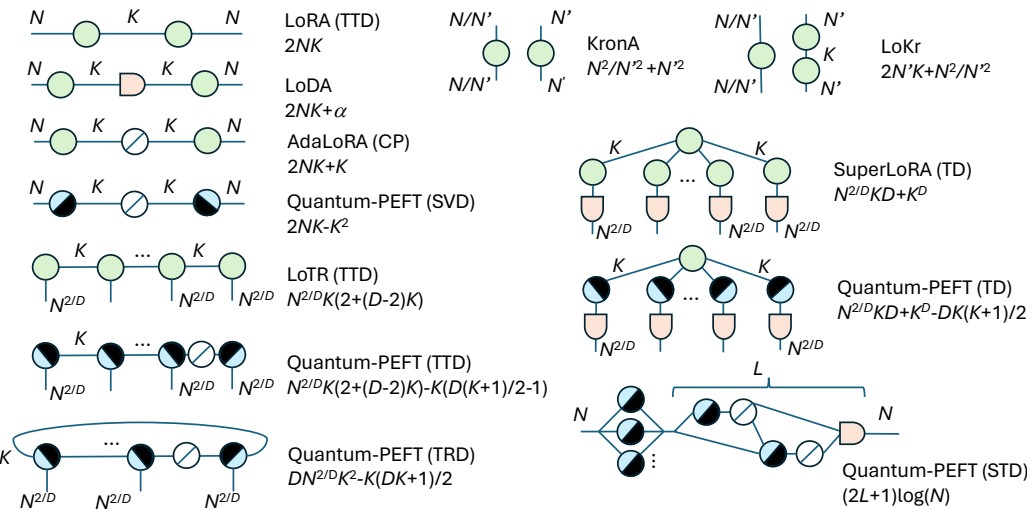

Figure 5: Tensor diagrams of Quantum-PEFT and LoRA variants in tensor network perspectives for a matrix size of $N$ and rank $K$. The number of parameters are also present. Circle denotes dense multi-linear tensor node. Slashed open circles denote diagonal node. Half-closed circles denote unitary node. Delay symbols denote nonlinear nodes.

### A.1  COMPARISONS OF DIVERSE UNITARY MAPPINGS

**Unitary mappings**  Various methods to generate unitary matrices from skew-symmetric matrices are possible. In Section 4.1, we focused on exponential and Taylor mappings. Given a skew-symmetric matrix $A = B - B^\top \in \mathbb{R}^{N' \times N'}$, we can generate a corresponding unitary (orthogonal) matrix, e.g., with exponential mapping, Cayley transform, Householder reflection, Givens rotation and those variants, respectively, as follows:

$$Q_{\mathsf{E}} = \exp(A), \quad Q_{\mathsf{C}} = (I + A)(I - A)^{-1}, \quad Q_{\mathsf{H}} = \prod_{k=1}^{K} \left( I - 2\,\mathfrak{N}[B_{:,k}]\mathfrak{N}[B_{:,k}]^\top \right), \quad (5)$$

$$Q_{\mathsf{G}} = \prod_{k=1}^{K} \prod_{n=k+1}^{N} G_{n-k}(B_{n,k}), \quad Q_{\mathsf{T}} = \sum_{p=0}^{P} \frac{1}{p!} A^p, \quad Q_{\mathsf{N}} = (I + A) \sum_{p=0}^{P} A^p, \quad (6)$$

where $\mathfrak{N}[\cdot]$ is a normalization operator for canonical coset decomposition (CCD) (Cabrera et al., 2010), and $G_n(\theta)$ denotes the Givens matrix which is identity except that the $n$ and $(n+1)$-th diagonal block is replaced with RY rotation. The mappings of $Q_{\mathsf{T}}$ and $Q_{\mathsf{N}}$ are respectively approximated versions of $Q_{\mathsf{E}}$ and $Q_{\mathsf{C}}$ to avoid matrix exponentiation and inversion via Taylor series and Neumann series approximations up to a polynomial order $P$. Note that $Q_{\mathsf{P}}$, $Q_{\mathsf{E}}$ and $Q_{\mathsf{G}}$ are identical to RY at $N' = 2$.

**Comparison of unitary mappings**  Fig. 6 shows the comparison of different unitary mapping methods over different matrix size $N$ for a rank of $K = 4$. We examined the unitarity test and speed bench on RTX6000 GPU for forward and backward processing. The unitarity error measures an averaged $\ell_\infty$ norm of $\|QQ^\top - I\|_\infty$ over a batch size of 32 and 10 random seeds. The exponential mapping uses `torch.linalg.matrix_exp`, and matrix inversion for Cayley transform uses `torch.linalg.solve`. We assume $P = 18$ polynomial order for Taylor and Neumann series. It was found that Neumann series and exponential mapping become inaccurate as the matrix size is

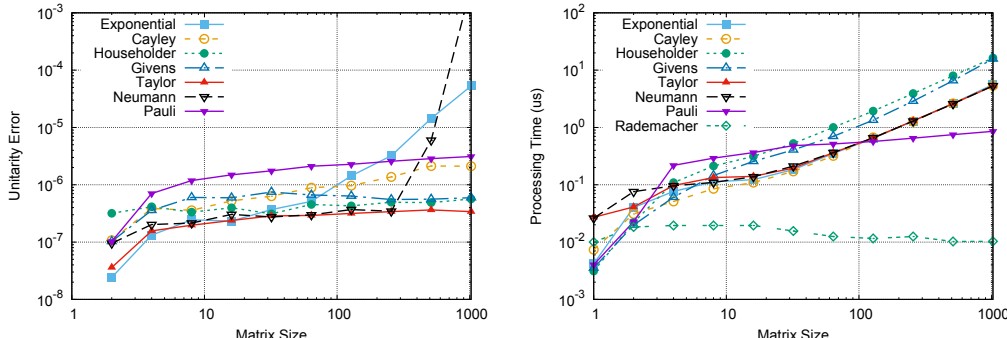

Figure 6: Unitarity error analysis and speed bench including forward and backward passes for different unitary mapping methods as a function of matrix size of $N$ for a rank of $K = 4$ on an NVIDIA RTX6000 GPU 24GB.

increased. While Pauli parameterization has relatively higher error than the rest of methods, it can be much faster in large matrix size. Householder reflections and Givens rotations had slower behaviors due to sequential nature. Although Rademacher diagonal matrix of $\{\pm 1\}^K$ has a low complexity and perfect unitarity (here, we used ReinMax trick), it alone does not cover the Stiefel manifold $\mathcal{V}_K(N)$. Overall, Taylor series method showed a good trade-off between accuracy and speed. Note that most large foundation models use thousands for a matrix size of $N$ per weight. Therefore, the accuracy and speed at large matrix size regimes are important. With these trade-offs in mind, in the experiments we evaluate the Taylor $Q_\mathsf{T}$ and Pauli $Q_\mathsf{P}$ parametrizations, where Pauli gives logarithmic number of trainable parameters in the ambient dimension and Taylor shows satisfactory speed for larger models.

## A.2 QUANTUM-INSPIRED PEFT MODULES

**Generalized measurements** As well as generalized-RY gates and CZ gates, we introduce generalized measurement module. Although quantum operation is linear, quantum measurement can be nonlinear in general. Hence, motivated from the quantum measurement to solve the linearity constraint, we can impose nonlinearity using activation functions. Using log-softmax after squaring corresponds to measuring quantum state probability. For our case, such nonlinear activations can be imposed at any mid-circuit operations. In Fig. 5, we introduce a new tensor diagram with delay symbols representing the nonlinear node. Nonlinear mapping can be also trainable when using another multi-layer perceptron (MLP) as used in LoDA. Letting $f(\cdot)$ be such a nonlinear function, tensor contraction can be done via *nonlinear* Einstein sum: $f^{\mathrm{out}}(\sum f^{\mathrm{in}}(\prod Q_{i,j}^{[k]}))$ for parent tensor nodes $\{Q^{[k]}\}$, where $f^{\mathrm{out}}$ and $f^{\mathrm{in}}$ denote outer nonlinearity and inner nonlinearity, respectively. Note that the nonlinear nodes can only pass the data after tensor contraction from all ancestor nodes.

## A.3 TENSOR NETWORK IMPLICATION

Fig. 5 shows tensor diagrams for various LoRA variants. Our Quantum-PEFT framework can unify them with reduced number of parameters by exploiting trainable orthogonal nodes, trainable diagonal nodes, and trainable nonlinear nodes. As mentioned, LoRA uses 2-mode tensor train decomposition (TTD) which is also known as matrix product state (MPS) tensor network. However, LoRA is parameter redundant as any tensor network via TTD can be reformulated into its canonical form having multiple unitary nodes and only one diagonal node, resulting into fewer parameters. LoDA introduced the nonlinear node in tensor network. AdaLoRA is based on CP decomposition, which has parameter redundant. LoTR extends LoRA towards higher-mode TTD. SuperLoRA uses another tensor network based on higher-oder Tucker decomposition (TD), while nonlinear mapping is optionally introduced. In fact, TTD and TD can be normalized except one node (i.e., the canonical form), and hence our Quantum-PEFT based on the Lie algebra can eliminate the redundant parameters to improve the efficiency for LoRA, LoTR and SuperLoRA. Similarly, our framework provides parameter-efficient unitary nodes in most other tensor networks including tensor ring decomposition (TRD), hierarchical Tucker decompostion (HTD) a.k.a. tree tensor network

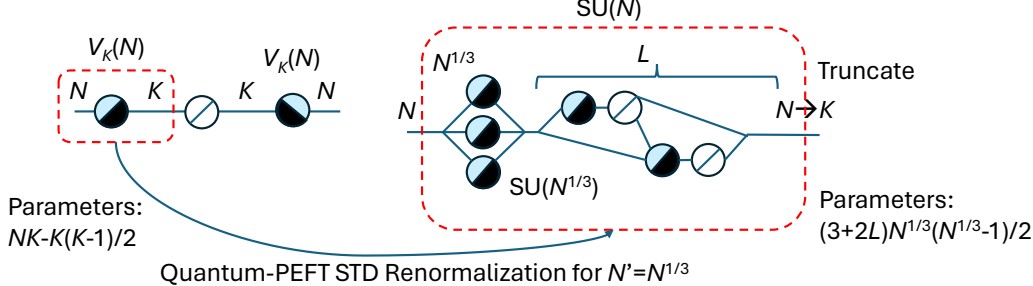

Figure 7: STD renormalization step example when $N' = N^{1/3}$. The total number of parameters is reduced from 729 to 180 for a unitary node when $K = 1, N = 3^6, N' = 3^2, L = 1$.

Table 9: Accuracy on the ViT CIFAR10 transfer learning task with varying entanglement layers $L$.

| $L$ | 1 | 2 | 3 | 4 | 8 | 16 | 32 |
|---|---|---|---|---|---|---|---|
| Accuracy | 96.09% | 96.54% | 96.71% | 96.71% | 96.71% | 96.71% | 96.71% |

(TTN), multi-scale entanglement renormalization ansatz (MERA), and projected entangled pair states (PEPS). As descussed, Pauli parameterization based on STD ansatz can further reduce the number of parameters for those tensor networks into a logarithmic scale. Note that STD parameterization can be regarded as a renormalization step of each orthogonal node in tensor networks. Fig. 7 shows an example of the STD renormalization step when $N$ is 3-folded into $N' = N^{1/3}$. The total number of parameters to represent the unitary node for $\mathcal{V}_K(N)$ can be reduced in a logarithmic order of $\log_{N'}(N)$. When $K = 1, N = 3^9, N' = 3^2, L = 1$, it becomes 180 from 729. Reducing the size of $N'$ can further improve the parameter efficiency.

Table 10 shows an example result for ViT CIFAR10 transfer learning task, using Taylor parameterization (with $K = K' = 4$ and $P = 18$) for different tensor networks, including CP, TRD, HTD (TTN), TD, and TTD (MPS). We find that all tensor networks offer competitive performance to LoRA.

## A.4 FURTHER ANALYSIS OF ENTANGLEMENT LAYERS

The relationship between circuit depth and entanglement capacity has been explored in quantum information theory, providing insights into the expressive power of quantum circuits. For instance, Sim et al. (2019) analyzed the entangling capabilities of various quantum circuits, establishing a connection between the number of layers $L$ and entanglement. It was shown that generally deeper circuits exhibit an increased entanglement capacity, which can contribute to richer representations in the context of quantum machine learning. In our proposed Quantum-PEFT framework, the number of alternating entanglement layers $L$ in the Pauli parametrization $Q_P$ (Equation (2)) governs the circuit depth. Deeper circuits, while potentially more expressive, they also introduce additional trainable parameters and computational overhead. Our empirical findings suggest that circuits with $L = 1$ provides a good balance between performance and efficiency for PEFT tasks. Increasing $L$ can lead to moderate performance improvements, but the gains tend to diminish with larger values of $L$, indicating a saturation effect. To further investigate the impact of $L$, we conducted a sensitivity analysis on the ViT CIFAR10 transfer learning task described in Section 5.4 We evaluated the performance of Quantum-PEFT across various values of $L$, while keeping other hyperparameters fixed and with the base ViT model quantized to 2 bits. The results, summarized in Table 9, demonstrate the saturation behavior, as no further gain is attained beyond $L = 3$. Overall, the optimal value of $L$ is task-dependent, depending on the complexity of the target task, where the trade-off between performance gains and increased computational complexity needs to be carefully considered.

Table 10: Different tensor network results with Taylor parameterization for ViT transfer learning from ImageNet-21k to CIFAR10. Base ViT is not quantized.

| Method | CP | TRD | HTD (TTN) | TD | TTD (MPS) |
|---|---|---|---|---|---|
| # Parameters | 0.074M | 0.147M | 0.026M | 0.074M | 0.111M |
| Accuracy | 98.53% | 98.14% | 98.11% | 98.05% | 98.81% |

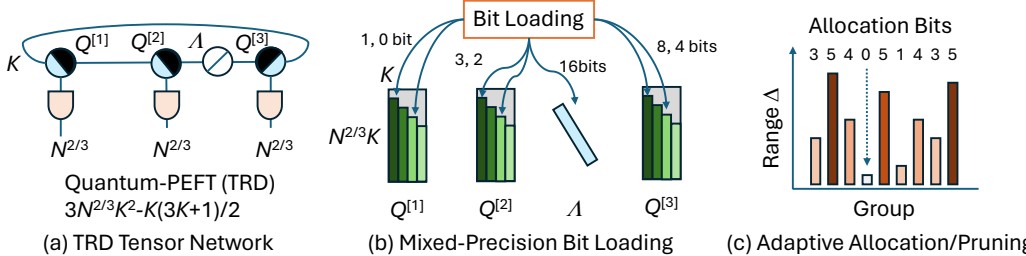

Figure 8: Mixed-precision Quantum-PEFT in 3-dimensional TRD tensor network. Each tensor node and tensor parameter can have non-uniform bit assignments. Adaptive bit loading depends on group range $\Delta$. Assignment of 0 bit corresponds to adaptive structural pruning.

## A.5 BROADER IMPACTS AND FUTURE WORK

It is interesting to investigate how we can further reduce the memory for trainable parameters by employing quantization or pruning.

**Mixed-precision tensor network** One could consider a mixed-precision tensor network, where each tensor node and its parameter group can have different precisions. Fig. 8(a) shows an example of Quantum-PEFT in 3-dimensional TRD tensor network. The TRD is formulated by 3 unitary nodes $\{Q^{[k]}\}$ and 1 diagonal node $\Lambda$. Specifically the $(i, j, k)$-th element is given by nonlinear Einstein sum: $W_{i,j,k} = f^{\text{out}}(\sum_{l,m,n} f^{\text{in}}(Q^{[1]}_{l,i,m} Q^{[2]}_{m,j,n} \Lambda_{n,n} Q^{[3]}_{n,k,l}))$. As shown in Fig. 8(b), each node has trainable parameters $\theta$, and we can adaptively assign more bits or fewer bits depending on the group range $\Delta_i = \theta_{i,\max} - \theta_{i,\min}$ for the $i$-th group. For example, the bit loading may use the following strategy: $q_i = \text{round}(q \log_2(\Delta_i^{\kappa}/\bar{\Delta}))$ with an average range $\bar{\Delta} = \mathbb{E}[\Delta_i^{\kappa}]$ where $q_i$ bits are assigned for the $i$-th group with an exponent $\kappa >= 0$. When $\kappa = 0$, it reduces to uniform bit loading: i.e., $q_i = q$ for all group $i$. More sophisticated but time-consuming strategy is to consider the quantization error of the weight matrix $\min|W_q - W|$, which requires combinatorial optimization.

When the bit allocation is zero (i.e., $\Delta_i$ is close to zero) as shown in Fig. 8(c), it corresponds to structural pruning except that the masked group can still hold non-zero values $\mu$. Further fine-grained pruning is also possible by nulling out $\theta$ if the value magnitude is smaller than a threshold. Therefore, it can accomplish an adaptive rank mechanism similar to AdaLoRA.

**Pretrained model compression** In fact, Quantum-PEFT framework can also be applicable to compress the pretrained model before adaptation. Tensor rank decomposition, quantization and pruning can be applied to pretrained model before transfer learning tasks, similar to Q-LoRA, R-LoDA, and S-LoDA. For ViT transfer learning task, we evaluated 3-bit quantization of pre-trained models.

## A.6 FURTHER COMPARISONS WITH RELATED WORK

We provide the following remarks elaborating on the difference between the proposed Quantum-PEFT method with recent related works on PEFT. Several recent works explore alternative approaches to parameter-efficient fine-tuning. Quantum-PEFT is a new technique for low-rank based PEFT. Some recent works explore alternative approaches to LoRA-based fine-tuning. For example, Pan

et al. (2024) fine-tune some important sampled layers while freezing the rest for some certain iterations, an orthogonal approach to LoRA. Their results show that Pan et al. (2024) share similar memory requirements as LoRA; in contrast, Quantum-PEFT achieves significantly higher parameter efficiency than LoRA. Other works adapt the intermediate embeddings learned by the models, which differs from LoRA-based methods that adapt the weights directly. In this sense, Wu et al. (2024) intervene on a low-rank subspace of the intermediate model embeddings rather than model weights. One disadvantage of this method is that it creates multiple additional hyperparameters, i.e., prefix and suffix positions to intervene on and which layers to intervene on, creating a combinatorial growth of hyperparameters. Therefore, their claimed higher efficiency comes at the cost of higher computational burden of hyperparameter tuning for each task. On the other hand, Quantum-PEFT mainly considers two hyperparameters, i.e., the intrinsic rank $K'$ and the number of entanglement layers $L$. Quantum-PEFT considers highly-efficient unitary quantum paramatrization, which is effectively full-rank. Recent work (Jiang et al., 2024b) has investigated high-rank fine-tuning. It employs a learnable square matrix $M \in \mathbb{R}^{\hat{K} \times \hat{K}}$ for full-rank fine-tuning and compatibility mappings to ensure that the dimensions match with the weight $W \in \mathbb{R}^{N \times M}$. They set $\hat{K} = \lfloor \sqrt{(N + M)K} \rfloor$ to achieve the highest rank with square matrix at the same total number of trainable parameters of LoRA with rank $K$. (Jiang et al., 2024b) is therefore not able to have a lower-than-LoRA scaling, which is instead achieved by Quantum-PEFT. (Chen et al., 2024b) is a tensor-network-inspired parameterization without consideration of unitary gain, where their parameterization leads to parameter redundancy. Our Pauli parameterization under Lie algebra can strictly maintain unitary constraint without parameter redundancy. In fact, (Chen et al., 2024b) is based on tensor folding, requiring in principle that the matrix size is factorizable as $d = d_1 \times d_2 \times \cdots \times d_N$. Therefore, it is not readily compatible for arbitrary size. For example, if the matrix size is $d = 257$, it is difficult to fold into multiple axis. We clearly provided the way to solve this issue by quantum Shannon decomposition, which enables to decompose into sum of powers-of-two: i.e., $N_1 = 256$ and $N_2 = 1$. Furthermore, Quantum-PEFT provides more general applicability and insight to any arbitrary tensor network to reduce the parameter number as shown in Figures 5 and 7, and provides more flexibility with adjustable entangling layer size $L$, which is not explored in (Chen et al., 2024b).

### A.7 LIST OF SYMBOLS

For ease of reference, we provide a table of notations used in this work in Table 11.

## B DETAILED EXPERIMENTAL SETUPS

### B.1 GLUE BENCHMARK

Below, we provide a summary of the tasks in the GLUE benchmark that are used in this work.

- SST-2: stands for The Stanford Sentiment Treebank, a dataset on sentiment analysis tasks with two labels. The size of the training set is 67k, and the size of the test set is 1.8k.

- CoLA, represents The Corpus of Linguistic Acceptability, a dataset on sentence classification with two labels. It consists of 8.5k training data and 1k test data.

- RTE: stands for The Recognizing Textual Entailment, including 2.5k training data points and 3k test data points.

- MRPC: represents The Microsoft Research Paraphrase Corpus, a dataset on pairwise text classification with 3.7k training points and 1.7k test points.

- STS-B: represents The Semantic Textual Similarity Benchmark, a task on measuring text similarity with 7k training points and 1.4k test points.

We select the same number of epochs for Quantum-PEFT as in AdaLoRA. We perform a hyper-parameters sweep for the learning rate over $\{0.01, 0.03, 0.06, 0.001, 0003, 0.006\}$. We select the best learning rate and the best checkpoints over each epoch. We present the hyperparameters for Quantum-PEFT in Table 12.

Table 11: List of symbols.

| Notation | Description |
|----------|-------------|
| $\mathrm{SU}(N)$ | Special unitary group of size $N$ |
| $\mathfrak{su}(N)$ | Lie algebra of $\mathrm{SU}(N)$ |
| $\mathrm{SO}(N)$ | Special orthogonal group of size $N$ |
| $\mathrm{O}(N)$ | Orthogonal group of size $N$ |
| $\mathcal{V}_K(N)$ | Stiefel manifold of $K$ orthonormal frames in $\mathbb{R}^N$ |
| $I_N$ | Identity matrix of size $N$ |
| $\mathbb{R}$ | Field of real numbers |
| $\otimes$ | Kronecker product |
| $[\cdot]^\top$ | Transpose |
| $\jmath$ | Imaginary number |
| $A_{m:n,:}$ | Submatrix of $A$ with rows $m$ to $n$ |
| $A_{:,k}$ | $k$-th column of matrix $A$ |
| $\mathrm{diag}[\cdot]$ | Creates a diagonal matrix |
| $L$ | Number of alternating entanglement layers |
| $q$ | Number of qubits |
| $N'$ | Orthogonal node size |
| $K'$ | Intrinsic rank |
| $P$ | Taylor expansion order |
| $Q_\mathsf{P}$ | Pauli-parameterized unitary matrix |
| $Q'_\mathsf{E}$ | Unitary matrix from exponential mapping |
| $Q'_\mathsf{T}$ | Unitary matrix from Taylor series expansion |
| $\mathsf{RY}(\theta)$ | Quantum RY rotation gate with angle $\theta$ |
| $\mathsf{CZ}$ | Quantum controlled-Z gate |
| $W$ | Pre-trained weight matrix |
| $\Delta W$ | Weight update matrix |
| $U$ | Left singular vector matrix |
| $V$ | Right singular vector matrix |
| $\Lambda$ | Diagonal matrix of singular values |
| $B, B_K$ | Lower triangular matrix and its parameter matrix |
| $C, S$ | Cosine and sine diagonal matrices from CSD |
| $n$ | Number of bits for quantization |
| $g$ | Quantization group size |
| $\beta, \mu$ | Quantization scale and zero-point |
| $\theta$ | Trainable parameter |
| $\theta_q$ | Quantized trainable parameter |

Table 12: Hyperparameter configurations for Quantum-PEFT on the GLUE benchmark.

| Hyperparameter | SST-2 | CoLA | RTE | MRPC | STS-B |
|----------------|-------|------|-----|------|-------|
| # GPUs | 1 | 1 | 1 | 1 | 1 |
| Optimizer | AdamW | AdamW | AdamW | AdamW | AdamW |
| Learning Rate Schedule | Linear | Linear | Linear | Linear | Linear |
| Weight Decay | 0.01 | 0.01 | 0.01 | 0.01 | 0.01 |
| Batch Size | 256 | 128 | 128 | 128 | 128 |
| Epochs | 24 | 25 | 50 | 30 | 25 |
| Warmup ratio | 0.1 | 0.1 | 0.1 | 0.1 | 0.1 |
| Max sequence length | 128 | 64 | 320 | 320 | 128 |
| Rank $K$ | 3 | 3 | 3 | 3 | 3 |
| $\alpha$ | 32 | 32 | 32 | 32 | 32 |
| Learning Rate | 0.006 | 0.01 | 0.06 | 0.01 | 0.03 |
| Unitary Parametrization | $Q_\mathsf{P}$ ($L=1$) | $Q_\mathsf{P}$ ($L=1$) | $Q_\mathsf{P}$ ($L=1$) | $Q_\mathsf{P}$ ($L=1$) | $Q_\mathsf{P}$ ($L=1$) |

## B.2   E2E BENCHMARK

Table 14 lists hyperparameters for the experiment on transfer learning task of E2E benchmark.

Table 13: CIFAR-10 transfer learning for ViT.

| Hyperparameter | LoRA | Quantum-PEFT |
|---|---|---|
| # GPUs | 1 | 1 |
| Optimizer | AdamW | AdamW |
| Learning Rate Schedule | Constant | Constant |
| Weight Decay | 0.01 | 0.01 |
| Batch Size | 32 | 32 |
| Epochs | 100 | 100 |
| Patience | 5 | 5 |
| Rank $K$ | 1,2,4 | 1, 4 |
| Learning Rate | 0.001 | 0.003 |
| Unitary Parametrization | — | $Q_\mathsf{P}$ ($L = 1$), $Q_\mathsf{T}$ ($P = 18$) |

Table 14: E2E benchmark for GPT2 Medium.

| Hyperparameter | LoRA | Quantum-PEFT |
|---|---|---|
| # GPUs | 4 | 4 |
| Optimizer | AdamW | AdamW |
| Learning Rate Schedule | Linear | Linear |
| Weight Decay | 0.01 | 0.01 |
| Batch Size | 8 | 8 |
| Epochs | 5 | 5 |
| Warmup Steps | 500 | 500 |
| Label Smooth | 0.1 | 0.1 |
| Rank $K$ | 4 | $2$ ($K' = 1$) |
| $\alpha$ | 32 | 32 |
| Learning Rate | 0.0002 | 0.002 |
| Unitary Parametrization | — | $Q_\mathsf{T}$ ($P = 3$) |

## B.3 VIT CIFAR10 TASK

Table 13 lists hyperparameters for the experiment on transfer learning task of ViT. The base ViT model (`google/vit-base-patch16-224`)[1] pretrained on ImageNet-21k has 12 layers of multi-head attention modules, each of which has 12 heads, 768 features, and a token length of 769. CIFAR10 is an image classification dataset having 10 classes of $32 \times 32$ colored images with 50k training samples and 10k test samples. We use up-sampling to $224 \times 224$ resolutions with random resized cropping and horizontal flip. The original classifier head has 1000 class output, and we selected 10 outputs based on the prediction score of CIFAR10 training data in prior to PEFT process. All weights and biases of the base ViT model including the classifier head are frozen after being quantized with 3-bit integers via rounding as described in Appendix A.5. Therefore, the base model is compressed from floating-point 32 bits to integer 3 bits (with auxiliary scale and zero values $\beta$ and $\mu$ for $g = 128$ group), i.e., from 330MiB to 34MiB storage. It was confirmed that less than 3-bit quantization for the base ViT model compression had poor performance: $56.0\%$ accuracy with 1 bit and $97.4\%$ with 2 bits. The required run-time on GPU A40 40GB was about 3.37 second per iteration, and 5284.16 second per epoch.

---

[1] https://huggingface.co/google/vit-base-patch16-224

