# OpenReview forum: "Quantum-PEFT: Ultra parameter-efficient fine-tuning"
_ICLR.cc/2025/Conference — ICLR 2025 Poster_

### Official Review · Reviewer_1BuM · 2024-10-22

**Soundness:** 3
**Presentation:** 3
**Contribution:** 3
**Rating:** 6
**Confidence:** 4

**Summary:**

This paper proposes a new formulation for low-rank based PEFT, based on quantum mechanical notation.
The authors demonstrate that their ansatz using Pauli Parametrization of low-rank adapters provides a superior paramters-accuracy ratio in numerious contemprary benchmarks

**Strengths:**

I find the authors approach of unification of various low-rank adapter notations based upon quantum-mechanical notations interesting.
The paper has strong numerical evaluations.

**Weaknesses:**

While I enjoyed reading most of the paper, a few things are missing in the explanation of the method, causing some neccessary revisions.

- The paper describe the compute ansatz in eq. (2), although the notations is quite heavy for readers that are not familiar with the field. The ansatz governs the compute instructions of the forward pass of the neural network. However, the backward-pass, especially computing the gradient update of the parameters is not covered. Especially with the dense notation, an algorithmic description is nessecary to explain how the gradient update is performed and a discussion of its computational feasibility should be added.

- Updating a tensor-factorization as Tensor-Trains or Tucker Decomposition naively may induces unexpected perils for the low-rank optimization. In particular (in combination with the lacking update scheme), it needs to be discussed if a gradient descend on the Factors of the proposed parametrization indeed decreases the overall loss function of the method.

**Questions:**

- How does your factorized update scheme relates to robust Riemannian optimization schemes on Stiefel Manifolds, e.g. as described in
[1] for Tucker Tensors and in [2,3] for Matrix Factorizations of the Type USV (as in Adalora)?

- Is there an option to make the method rank-adaptive, as e.g. in [1,2,3] or AdaLora?

[1] Emanuele Zangrando, Steffen Schotthöfer, Gianluca Ceruti, Jonas Kusch, and Francesco Tudisco.
Rank-adaptive spectral pruning of convolutional layers during training. In Advances in Neural
Information Processing Systems, 2024.

[2] Steffen Schotthöfer, Emanuele Zangrando, Jonas Kusch, Gianluca Ceruti, and Francesco
Tudisco. Low-rank lottery tickets: finding efficient low-rank neural networks via matrix differential equations. In Advances in Neural Information Processing Systems,
2022. URL https://proceedings.neurips.cc/paper_files/paper/2022/
file/7e98b00eeafcdaeb0c5661fb9355be3a-Paper-Conference.pdf.

[3] Steffen Schotthöfer and M. Paul Laiu. Federated dynamical low-rank training with global loss
convergence guarantees, 2024. URL https://arxiv.org/abs/2406.17887.

---

> ### Author Response · Authors · 2024-11-22
>
> We thank the reviewer for the valuable feedback and address the concerns below.
>
> > Q1. The paper describe the compute ansatz in (2). However, the backward-pass, especially computing the gradient update of the parameters is not covered
>
> Thank you for your comment. The trainable parameter $\theta_{k,i}$ in Eq. (2) is associated with individual RY gate, defined in (1). RY gate is based on trigonometric function wrt $\theta$, and thus closed-form expression of the gradient is known:
> $\frac{\partial \mathsf{RY}(\theta)}{\partial \theta} = \frac{1}{2}
> \begin{bmatrix}
> -\sin(\theta/2) & -\cos(\theta/2) \\\\
> \cos(\theta/2) & -\sin(\theta/2) \\\\
> \end{bmatrix}$.
> Eq. (2) is based on Kronecker product of RY gates: $\otimes_k RY(\theta_{k,i})$. Its derivative is straightforward.
> Specifically, we have
> $\frac{\partial}{\partial \theta_{m,i}} \bigotimes_{k=1}^q RY(\theta_{k,i})= RY(\theta_{1,i}) \otimes RY(\theta_{2,i}) \otimes \cdots \otimes \frac{\partial RY(\theta_{m,i})}{\partial \theta_{m,i}} \otimes RY(\theta_{m+1,i}) \otimes \cdots$.
> Correspondingly, the gradient update of quantum parameters is computationally feasible.
> In experiments, we simply used PyTorch's autograd.
> We leave further engineering optimization as a future work.
> Nevertheless, our analysis in Fig. 6 shows that the gradient update of the Pauli parameterization is still competitive even without custom kernels.
> In the revised manuscript, we added a discussion on how we compute the gradient update.
>
> > Q2. A gradient descend on the Factors of the proposed parametrization
>
> Indeed, the convergence of quantum tensor networks and quantum machine learning is not fully understood yet.
> Nevertheless, there are many empirical literature showing sufficient trends of loss decreasing in many practical cases, e.g., [5].
> Remarkably, [4] provides a theoretical proof of sublinear convergence under a reasonable assumption.
> In our case, we empirically observed no issues in loss convergence and added discussion on the employed gradient method in Sec. 4.1.
>
> > Q3. How does your factorized update scheme relates to [1,2,3]?
>
> [2] optimizes the neural network weight matrices with gradient flow along the low rank weight manifold. [3] considers for manifold-constrained optimization in the federated learning setting. [1] proposes rank-adaptivity in CNNs via low-rank Tucker.
> Quantum-PEFT parameterizes $U, V$ factors of low-rank delta weights through trainable intrinsic parameters instead of directly optimizing the low-rank factors of network weights.
> This approach differs from [1, 2, 3], which employ manifold optimization to maintain feasibility of the factors.
> In Quantum-PEFT, the $U,V$ matrices are not directly trainable parameters, but rather computed via efficient Pauli unitary parameterization from underlying smaller learnable parameters. Hence, our optimization happens in the Euclidean space of the parameters of the parameterization.
> We included discussions wrt [1,2,3] in Sec. 2.
>
> > Q4. Is there an option to make the method rank-adaptive?
>
> In our current method, we introduced an intrinsic rank $K'$ as a mechanism to further control the effective number of trainable parameters compared to the specified rank $K$ by keeping the top $K'$ columns of Lie parameters, while the other parameters are frozen or null-out. By choosing $K' < K$, we control the intrinsic dimensionality of the learned subspace, offering a finer-grained control and enabling higher parameter reduction. As demonstrated in our sensitivity analysis for $K'$ (Table 8), reducing $K'$ can significantly diminish the number of parameters while incurring minimal performance degradation. For example, even with $K'$ reduced to 1, the performance drop is only 0.49%, while still maintaining a competitive advantage over LoRA.
>
> While the current framework allows for manual adjustment of $K'$ as shown in the sensitivity analysis, one could devise a fully rank-adaptive method that dynamically adjusts $K'$ during training.
> A crucial aspect of such an adaptive strategy would be developing a principled approach for ranking the importance of the Lie algebra parameters.
> In fact, unlike AdaLoRA, where the singular values directly correspond to the importance of each rank component, the relationship is not as straightforward in Quantum-PEFT due to our Pauli parameterization. In future work, it would be interesting to rigorously analyze this relationship and incorporate it into a $K'$-adaptive algorithm.
>
> [1] Rank-adaptive spectral pruning of convolutional layers during training. Neurips 2024.
> [2] Low-rank lottery tickets: finding efficient low-rank neural networks via matrix differential equations. Neurips 2022.
> [3] Federated dynamical low-rank training with global loss convergence guarantees, 2024.
> [4] Analyzing convergence in quantum neural networks: Deviations from neural tangent kernels. ICML 2023.
> [5] Improving gradient methods via coordinate transformations: Applications to quantum machine learning. Physical Review Research.

---

> > ### Comment · Reviewer_1BuM · 2024-11-27
> >
> > Thank you for your clarifying comments.
> >
> > I'm curious,  how does the Pauli parametrization relate to the Tensor-Train format? The paragraph in Line 320 mentions that LoRA can be interpreted as a 2-mode TT, but can Pauli parametrization be interpreted as an higher order tensor train?

---

> > > ### Author Response · Authors · 2024-11-29
> > >
> > > The relation of the Pauli parameterization to the Tensor-train is illustrated in Fig. 5.
> > > LoRA is 2-mode TTD (denoted as ''LoRA (TTD)'' in Fig. 5), which has 2 tensor nodes having $2NK$ parameters in total.
> > > It is well-known that any tensor trains can be transformed into a canonical form, where all tensor nodes are unitary with one additional diagonal node.
> > > The canonical form of LoRA (2-mode TTD) becomes an SVD form, having left unitary, right unitary, and diagonal, which is denoted as ''Quantum-PEFT (SVD)'' in Fig. 5. This has benefits from non-canonical form: 1) the required number of parameters is reduced from $2NK$ to $2NK-K^2$; 2) unitary nodes have a numerical stability preventing magnitude explosion.
> > >
> > > This canonical form is important in parameter efficiency especially for higher ranks. For example, if rank is full $K=N$, then LoRA of size $2NK=2N^2$ has 2-times more parameters than the original weight matrix of size $N^2$, and thus LoRA is parameter-redundant for above half ranks $K\geq N/2$. Whereas the canonical form has no redundancy having the same number of parameters $2NK-K^2=N^2$ even for full ranks.
> > >
> > > Higher-mode tensor trains extend LoRA to higher orders, e.g., denoted as ''LoTR (TTD)'' in Fig. 5. Our framework provides a way to improve basic LoTR by reformulating into its canonical form (denoted as ''Quantum-PEFT (TTD)'' in Fig. 5), which offers further parameter efficiency and stability.
> > >
> > > The Pauli parameterization achieves enhanced parameter efficiency through tensor renormalization, where each tensor node is further decomposed into multiple unitary nodes and diagonal nodes. As mentioned above, any arbitrary tensor network can be formulated into the canonical form with unitary and diagonal nodes. Those unitary nodes can be further decomposed into $N'\times N'$ unitary matrices in Pauli representations (denoted as ''Quantum-PEFT (STD)'' in Fig. 5), which can further reduce the number of parameters in a logarithmic scale. The tensor renormalization with Pauli parameterization is explained in Fig. 7.
> > >
> > > In summary, Quantum-PEFT is a framework to improve the parameter efficiency by imposing unitary nodes in transforming into its canonical form for arbitrary tensor networks (not only 2-mode tensor trains or higher-order tensor trains). In addition, Quantum-PEFT offers a way of tensor renormalization to further improve the efficiency to represent unitary nodes. We thank the reviewer for the question and we remain available for further discussions.

---

> ### Comment · Area_Chair_1MmG · 2024-11-27
>
> Dear Reviewer,
>
> The authors have provided their rebuttal to your comments/questions. Given that we are not far from the end of author-reviewer discussions, it will be very helpful if you can take a look at their rebuttal and provide any further comments. Even if you do not have further comments, please also confirm that you have read the rebuttal. Thanks!
>
> Best wishes,
> AC

---

> ### Comment · Reviewer_1BuM · 2024-11-30
>
> I thank the authors for their clarification. I decide to keep my score and raise my confidence score.

---

### Official Review · Reviewer_og4k · 2024-11-02

**Soundness:** 3
**Presentation:** 3
**Contribution:** 3
**Rating:** 6
**Confidence:** 3

**Summary:**

This paper proposes Quantum-PEFT, a novel parameter-efficient fine-tuning method inspired by quantum computing concepts. The core contribution lies in achieving logarithmic parameter scaling through Pauli parameterization while maintaining orthogonality via Stiefel manifold mapping.

**Strengths:**

- Novel theoretical framework combining quantum-inspired parameterization with PEFT
- Significant reduction in parameter count compared to existing methods
- Comprehensive experiments across multiple tasks and architectures

**Weaknesses:**

1. The paper lacks clear delineation of learnable parameters in the mathematical formulations. While extensive comparisons with LoRA are provided, the fundamental step of identifying and justifying which parameters are learnable is overlooked. This impedes understanding of the method's core mechanism.

2. While the parameter efficiency is well-demonstrated, the computational complexity analysis is insufficient:
- The introduction of Stiefel manifold mapping introduces additional computational operations beyond standard matrix arithmetic
- Limited discussion of practical computational bottlenecks
- Experimental results show no significant advantage in fine-tuning time efficiency
- Absence of detailed analysis on computational overhead

**Questions:**

* In Figure 5, which provides Intuitive illustrations of idea of Q-PEFT, is relegated to the appendix. Given its importance for understanding the method's foundations, especially for researchers from non-quantum backgrounds, innovations within this figure should be moved to the main text.

* The mathematical notation requires better organization and clearer presentation. A dedicated notation section would significantly improve readability and accessibility.

* What are the practical limitations and deployment considerations for real-world applications?

---

> ### Author Response · Authors · 2024-11-22
>
> We thank the reviewer for the valuable feedback.
>
> > Q1. Delineation of learnable parameters
>
> We have carefully revised Section 4.1 to more clearly delineate the trainable parameters. In Quantum-PEFT, we use parameter-efficient representation of unitary transformations, i.e., the trainable parameters are Lie algebra for unitary parameterization.
> Specifically, in Eq. (2), the trainable parameters are the rotation angles $\theta_{k,l}$ associated with the employed RY gates.
> In fact, in (2) the Lie algebra, i.e., $\theta_{k,l}$, for each RY gates is trainable.
> The RY gate has a single parameter, as in (1), and thus the total number of trainable parameters for (2) is $(2L+1)\log_2(N)-2L$.
> This logarithmic scaling is a key advantage of our approach.
>
> As an example, for 5-qubit case with $N=2^5$, we illustrated it in Fig. 2(a): the left-most 5 RY gates in Fig. 2(a) correspond to the right-most RY term in (2); the middle 4 RY gates in Fig. 2 correspond to the middle term in (2); the last 4 RY gates in Fig. 2 correspond to the first RY term in (2).
> For more general cases of $N'>2$ in Fig. 2(b), the lie algebra, i.e., strictly lower triangular matrix $B$, has $K'N'-K'(K'+1)/2$ parameters per generalized RY gate as in Fig. 3(a).
> Besides orthogonal nodes, diagonal node has additional trainable parameters of size $K$ as in Fig. 3(b).
>
> > Q2. Computational complexity analysis
>
> For the Pauli parameterization $Q_\mathsf{P}$, we obtain better-than-LoRA parameter scaling with runtime $O[(L+1) N\log_2(N)]$, using the efficient Kronecker Shuffle algorithm (Plateau, 1985), where in the experiments we used $L=1$. LoRA has runtime $O(2NK)$. Detailed analysis of computational complexities is given in *Computational complexity* paragraph in Sec. 4.2, where we have emphasized that our parameterizations remain within the same complexity as existing methods. In summary, while the constant factors may slightly differ, this demonstrates that the overall complexity remains highly competitive with LoRA even with the mapping.
>
> Wall-clock time is measured in Table 4 and reported below for reference, where we compare Quantum-PEFT against baselines when fine-tuning GPT2.
>
> **Table og4k-1**: Efficiency comparison on GPT2.
> | Resource|LoRA|AdaLoRA|Quantum-PEFT|
> |-|-|-|-|
> |Training Time (ms/batch)|1719.06|1795.91|1723.39|
> |Memory Ratio|4.03×|4.03x|1×|
>
> We find that **all methods have very similar runtime**; notably, we observe a **4.03x reduction in number of parameters with Quantum-PEFT** and overall better performance than LoRA with 4x fewer parameters (see Table 3 in manuscript). Our implementation is based on Kronecker product operation and optimizations with custom CUDA kernels are possible.
> We made a rigorous runtime analysis for different unitary parameterizations in Fig. 6, which shows Pauli can be faster than other parameterizations at large sizes.
>
> > Q3. Fig. 5 placement
>
> We agree that Fig. 5 gives an intuitive illustration of the core concepts in Quantum-PEFT to benefit a broader audience. It illustrates the structure of the quantum circuit from its fundamental building quantum gates as unitary matrix transformations from small unitary matrices. Therefore, we have moved it to the main body as Fig. 2 for clearer understanding of Eq. (2) for all readers. We have also revised Introduction and Preliminaries sections with more detailed and clearer self-contained explanation of the necessary background to improve accessibility to researchers from different backgrounds.
>
> > Q4. A dedicated notation section would significantly improve readability and accessibility
>
> We have substantially expanded the notation paragraph at the beginning of Sec. 3 based on your feedback.
> We have also added a table defining all symbols used in the paper (see Appendix A.7), serving as a quick reference for the reader. We believe these enhancements significantly improve the accessibility and readability of the presentation.
>
> > Q5. What are the practical limitations and deployment considerations?
>
> Quantum-PEFT offers significant advantages in parameter efficiency for fine-tuning, achieving significant parameter reduction compared to LoRA. In implementation, one needs to consider: i) redundant memory to generate full-rank unitary matrix;  ii) algorithmic performance for unitary mapping, and iii) numerical error at low-precision.
> In our work, we provided solutions for i) by using tensor contraction ordering (Pfeifer et al.), ensuring memory efficiency and avoiding redundant storage of full unitary matrices, and ii) by Kronecker Shuffle algorithm (Plateau), performing a detailed experiment for the required complexity and unitarity accuracy in Fig. 6, showing that the Pauli parameterization offers competitive speed at little unitarity error compared to other unitary parametrizations. Regarding iii) w.r.t. the impact with low-precision quantization, we discussed quantization-aware training in Appendix A.5 and Table 7, showing promising results even with 1-bit quantization.

---

> > ### Comment · Reviewer_og4k · 2024-12-01
> >
> > Thank you for your revision and experiments. The running time is much better than I think. Considering the contributions of this work, I will remain the previous score.

---

> ### Comment · Area_Chair_1MmG · 2024-11-27
>
> Dear Reviewer,
>
> The authors have provided their rebuttal to your comments/questions. Given that we are not far from the end of author-reviewer discussions, it will be very helpful if you can take a look at their rebuttal and provide any further comments. Even if you do not have further comments, please also confirm that you have read the rebuttal. Thanks!
>
> Best wishes,
> AC

---

### Official Review · Reviewer_XPHL · 2024-11-03

**Soundness:** 3
**Presentation:** 3
**Contribution:** 2
**Rating:** 6
**Confidence:** 3

**Summary:**

This paper introduces Quantum-PEFT, a novel parameter-efficient fine-tuning method that uses quantum-inspired unitary parameterizations. The key innovation is using Pauli parameterization to achieve logarithmic scaling of trainable parameters with matrix dimensions, compared to linear scaling in traditional LoRA methods. The method is evaluated on language and vision tasks, demonstrating comparable performance to LoRA while using significantly fewer parameters.

**Strengths:**

1. The paper introduces a parameterization based on quantum circuits that require only (2L+1)log₂(N)-2L parameters, a significant improvement over LoRA's 2NK parameters.
2. The reduction in parameters is experimentally verified.
3. The authors proposed generalized RY and CZ quantum gates for arbitrary dimensions beyond power-of-2.
4. The proposed method seems robust under quantization.

**Weaknesses:**

1. Analysis of how L layers affect entanglement capacity is limited. There is no evaluation of entanglement entropy between layers.

2. The paper only focuses on benchmarking against LoRA/adapter variants on small models (such as GPT2). Benchmarks on larger (and newer) models such as LLaMA seem to be quite standard and more related to practical use cases.

1. Some of the recently proposed PEFT methods are not adequately addressed/discussed (at least mentioned somewhere in the paper), e.g.

[1] https://arxiv.org/abs/2403.17919 (NeurIPS)

[2] https://arxiv.org/abs/2404.03592 (NeurIPS)

[3] https://arxiv.org/abs/2405.12130

[4] https://arxiv.org/abs/2406.00132 (NeurIPS)

Both [3] and [4] seem to be able to model high-rank matrices, and [4] appears to be also based on quantum circuit.

**Questions:**

1. How does Eq. 4 work?
2. What is the optimal number of alternating entanglement layers L for different model scales?
3. For the quantum Shannon decomposition approach to handle non-power-of-two dimensions, how does the decomposition choice (N₁, N₂) affect model performance?
4. Does the proposed method have a large computational overhead from the entanglement layers?
5. How does the proposed method compare with other methods mentioned above? My understanding of the main difference between this method and [4] is the use of unitary and diagonal matrices. Is this correct?
6. The proposed method focuses on reduction in parameters. However, [2] and [4] claim both reduction in parameter and performance improvement. Does the proposed method use even fewer parameters?
6. How is the proposed method related to quantum machine learning?

---

> ### Author Response · Authors · 2024-11-22
> **Official Comment by Authors (Part 1/3)**
>
> We thank the reviewer for the insightful feedback and we address the remaining concerns below.
>
> ---
>
> > Q1. Analysis of how L layers affect entanglement capacity.
>
> Thank you for the comment. There is existing literature on the relationship between circuit depth and entanglement, e.g., [7] provided a rigorous analysis of entangling capacity with different quantum circuts. It was shown that generally quantum circuits exhibit increased capacity with more layers. In our experiments, a single entanglement layer ($L=1$) in the simplified two-design ansatz showed good performance for PEFT tasks, while increased layers can moderately improve the performance at the cost of increased parameters and computational overhead. We have included the discussion of the entangling layers in the revised manuscript (highlighted in blue) accordingly in Appendix A.4.
>
>
> Furthermore, we conducted a sensitivity analysis w.r.t. the number of entanglement layers $L$ in Table XPHL-1, showing the accuracy of the ViT CIFAR10 transfer learning task of Section 5.4 with a 2-bit quantized base ViT model, across different values of $L$.
> We observe a saturation effect, where the performance gains plateau beyond $L=3$.
> It is reasonable that the optimal $L$ is task-dependent, with more complex tasks potentially benefiting from a deeper circuit.
> We have included these findings in the revised manuscript in Table 9.
>
> **Table XPHL-1**: Accuracy over the number of entanglement layers $L$ in ViT CIFAR10 task. Base ViT is 2-bit quantized.
> |$L$ | 1 | 2 | 3 | 4 | 8 | 16 | 32 |
> |----|---|---|---|---|---|----|----|
> | Accuracy | 96.09 | 96.54 | 96.71 | 96.71 | 96.71 | 96.71 | 96.71 |
>
> ---
>
> > Q2. Benchmarks on larger and newer models.
>
> Following the reviewer’s suggestion, we have conducted additional experiments by fine-tuning Mistral-7B [5] on the standard GLUE benchmark. Mistral 7B is a recent LLM that has been shown to outperform Llama 2-13B on many benchmarks [5].
> We provide empirical comparisons below, where we used the same rank and alpha as in Section 5.1 and employed AdamW with 5 epochs and 4-bit quantization, and fine-tune query/key/value/gate projection matrices, following the setups of [6], for all methods.
> The results are presented in Table XPHL-2 with LoRA and AdaLoRA, which are the strongest baselines on GLUE in Section 5.1. Quantum-PEFT maintains its significant advantage in parameter efficiency even at 7B scale while achieving competitive performance with LoRA and AdaLoRA. We observed a **reduction in trainable parameters by a factor of 4.67x**, while even **exceeding the performance of LoRA and AdaLoRA** in almost all datasets. Quantum-PEFT offers a highly parameter-efficient alternative to LoRA-based approaches, which is crucial when scaling to billion-scale large language models.
> We have included these new experiments in Section 5.3 of the revised manuscript.
>
> **Table XPHL-2**: Results with Mistral-7B on GLUE Benchmark.
> | Method         | # Trainable Parameters | SST-2 | CoLA | RTE   | MRPC | STS-B | Avg. |
> |----------------|-------------------------|-------|------|-------|------|-------|------|
> | LoRA           | 3,538,944                      | 96.21   | 68.83   | 87.46 | 87.74   | 90.70   | 86.19   |
> | AdaLoRA        | 3,539,328                      | **96.90**   | 70.38   | 87.53 | 89.52   | 90.96    | 87.06   |
> | **Quantum-PEFT**   | **758,016**                      | 96.67    | **70.61**  | **88.10** | **89.94**   |  **91.63**  | **87.39**   |

---

> ### Author Response · Authors · 2024-11-22
> **Official Comment by Authors (Part 2/3)**
>
> > Q3. Comparative analysis with recent works [1,2,3,4].
>
> We first provide discussions and then present empirical comparisons.
>
> - Quantum-PEFT is a new technique for highly efficient reparameterization-based fine-tuning . [1] is orthogonal to LoRA-based techniques: it finetunes some important sampled layers while freezing the rest for some certain iterations. Their results show that [1] shares similar memory requirements as LoRA, while our method achieves significantly higher parameter efficiency than LoRA.
> - [2] intervenes on a low-rank subspace of the intermediate model embeddings rather than model weights. This creates additional hyperparameters, i.e., prefix and suffix positions to intervene on and which layers to intervene on, creating a combinatorial growth of hyperparameters. Therefore, their claimed higher efficiency comes at the cost of higher computational burden of hyperparameter tuning for each task. On the other hand, Quantum-PEFT mainly considers two hyperparameters, i.e., the intrinsic rank $K'$ and the number of entanglement layers $L$.
> We conduct a sensitivity analysis for $K'$ in Table 8, where it is shown that reducing $K'$ incurs in minimal performance degradation. For example, even with $K'$ reduced to 1, the performance drop is only 0.49%, while still maintaining a competitive advantage over LoRA.
> Additionally, unlike methods like Quantum-PEFT learning delta weights directly, their updates cannot be merged with the model weights, so their method incurs in computational overhead at inference time.
> - [3] employs a learnable square matrix $M \in \mathbb{R}^{\hat{K} \times \hat{K}}$ for full-rank fine-tuning and compatibility mappings to ensure that the dimensions match with the weight $W \in \mathbb{R}^{N \times M}$. They set $\hat{K}=\lfloor \sqrt{(N+M)K} \rfloor$ to achieve the highest rank with square matrix at the same total number of trainable parameters of LoRA with rank $K$. [3] is therefore not able to have a lower-than-LoRA scaling, which is instead achieved by Quantum-PEFT.
> - [4] is a tensor-network-inspired parameterization without consideration of unitary gain, so their parameterization leads to parameter redundancy and potential instability.
> Our Pauli parameterization under Lie algebra can strictly maintain unitary constraint without parameter redundancy.
> [4] is based on tensor folding, requiring in principle that the matrix size is factorizable as $d=d_1\times d_2 \times \cdots \times d_N$. Therefore, [4] is not readily compatible for arbitrary size. For example, if the matrix size is $d=257$, it is difficult to fold into multiple axis. We clearly provided the way to solve this issue by quantum Shannon decomposition, which enables to decompose into sum of powers-of-two: i.e., $N_1=256$ and $N_2=1$.
> Furthermore, our paper provides more general applicability and insight to any arbitrary tensor network to reduce the parameter number as shown in Figs. 5, 7, and provides more flexibility with adjustable entangling layer size, which is not explored in [4].
>
> We have included the above works in the Related Work section in main body and added detailed comparative discussions in Appendix A.6.
>
>
> To address the reviewer's concern, we have conducted additional experimental comparisons with the very recent MORA [3] and QuanTA [4, to appear in NeurIPS 2024], which are comparable to ours learning in the low-rank subspace of model weights.
> We follow the setups of Table 2 in the paper and use rank 3 for MORA as for the other methods.
> W.r.t. QuanTA [4], using their optimal setting $d=768=2^8*3$ to give the lowest number of parameters, **QuanTA [4] still has 0.093M trainable parameters compared to 0.013M parameters for our method**. We report the results in Table XPHL-3. We observe that Quantum-PEFT outperforms both MORA and QuanTA while using 37.87x and 7.15x less parameters, respectively.
>
> **Table XPHL-3**: Comparisons on GLUE with DeBERTaV3 including the very recent related works [3,4].
> | Method   | # Trainable Parameters | SST-2 | CoLA | RTE | MRPC | STS-B | Avg. | Memory |
> |----------|-------------------------|-------|------|-----|------|-------|------|--------|
> | FT       | 184M                    | 95.63 | 69.19| 83.75 | 89.46 | 91.60 | 85.93 | 14200× |
> | LoRA     | 0.33M                   | 94.95 | 68.71 | 85.56 | 89.71 | 91.68 | 86.12 | 25.38× |
> | MORA [3]    | 0.49M                      | 95.79    | 67.13 | 85.19   | 89.08   | 90.13    | 85.46 | 37.87x     |
> | QuanTA [4]    | 0.093M                      | 95.30    | 67.75 | 84.48   | 89.22   | 91.01    | 85.55  | 7.15x     |
> | **Quantum-PEFT**    | **0.013M**              | 95.85 | 67.85 | 86.57 | 90.78 | 91.06 | 86.42 | **1×** |

---

> ### Author Response · Authors · 2024-11-22
> **Official Comment by Authors (Part 3/3)**
>
> > Q4. How does Eq. 4 work?
>
> Eq. 4 demonstrates recursive cosine-sine decomposition (CSD) to construct a unitary matrix of arbitrary size by using multiple unitary matrices of smaller size. Tensor decomposition used in, e.g., QuanTA [4], is not readily compatible to arbitrary size of tensor. CSD overcomes this restriction. For example with $N=257=2^8+2^0$, CSD gives a way to use unitary representation with $N_1=256=2^8$ (i.e., 8-qubit) and $N_2=2^0$ (i.e., 0-qubit). 8-qubit matrices $U_1, V_2\in\mathbb{R}^{256\times 256}$ are easily parameterized by tensor decomposition, while 0-qubit special unitary is $U_2=V_1=1$.
> Eq. 4 becomes
> $$U=
> \begin{bmatrix}
> U_1 & 0_{256\times 3} \\\\
> 0_{3\times 256} & 1 \\\\
> \end{bmatrix}
> \begin{bmatrix}
> \cos(\theta) & -\sin(\theta) & 0_{1\times 256} \\\\
> 0_{256\times 1} & 0_{256\times 1} & I_{256} \\\\
> \sin(\theta) & \cos(\theta) & 0_{1\times 256} \\\\
> \end{bmatrix}
> \begin{bmatrix}
> 1 & 0_{1\times 256} \\\\
> 0_{256\times 1} & V_2 \\\\
> \end{bmatrix}.$$
> Similarly, arbitrary size of $N$ can be decomposed by representing its binary notation recursively, e.g., $259=2^8+2^1+2^0$, where CSD is sequentially applied: first CSD for $(N_1,N_2)=(2^8,2^1+2^0)$ and the second CSD for $(2^1, 2^0)$.
>
> ---
>
> > Q5. How does the decomposition choice (N₁, N₂) affect model performance?
>
> Thank you for the interesting question. Binarization of any number into largest to smallest power of 2 is a natural choice, e.g., with $N=2^8+2^6+2^2$, the first CSD may use $(N_1,N_2)=(2^8, 2^6+2^2)$, and the second CSD $(N_1,N_2)=(2^6, 2^2)$. This decomposition choice is sufficient to represent any unitary matrix and the number of parameters does not change if decomposed in a different way. For example, the first CSD could use $(N_1,N_2)=(2^8+2^2, 2^6)$, and the second CSD $(2^8, 2^2)$.
> The required number of parameters are exactly same, and the final matrix is equivalent.
> Although slight numerical difference may exist, we think the impact is very minor.
>
> ---
>
> > Q6. Does the proposed method have a large computational overhead from the entanglement layers?
>
> For the Pauli parameterization $Q_\mathsf{P}$, we obtain better-than-LoRA parameter scaling with runtime $\mathcal{O}[(L+1) N\log_2(N)]$, using the efficient Kronecker Shuffle algorithm (Plateau, 1985), where we used $L=1$ in the experiments. It scales linearly with entanglement layers. LoRA has runtime $\mathcal{O}(2NK)$. Detailed analysis of computational complexities is given in the *Computational complexity* paragraph in Section 4.2, where we have emphasized that our parameterizations remain within the same complexity as existing methods. In summary, while the constant factors may slightly differ, this demonstrates that the overall complexity remains highly competitive with LoRA even with the mapping. In our experiments in Table 4, we observed that all compared methods have very similar runtime, but Quantum-PEFT provides the largest memory savings compared to the other methods. We would like to further note that our current implementation is based on Kronecker product operation and further engineering optimizations with custom CUDA kernels may be possible.
>
> ---
>
> > Q7. Advantage in both parameter reduction and performance.
>
> Thank you for your suggestions, at the larger model size of Mistral-7B, **we have observed that we have both memory and performance advantage**, and we also perform better than very recent methods ([3,4]) that use more parameters in the DeBERTaV3 experiments, as shown in answers to Q2 and Q3.
>
> ---
>
> > Q8. How is the proposed method related to quantum machine learning?
>
> Our contribution lies in the innovative way we *translate* key concepts from quantum machine learning, i.e. efficient unitary parameterizations and quantum tensor network structures, into classical deep learning. To make the connections clearer, we have added Fig. 2 to the main text for clearer understanding of Eq. (2). Fig. 2 illustrates the structure of the quantum circuit from its fundamental building quantum gates as unitary matrix transformations from small unitary matrices. We have also revised Introduction and Preliminaries sections with more detailed and clearer self-contained explanation of the necessary background to improve accessibility to researchers from different backgrounds.
>
> ---
>
> [1] LISA: Layerwise Importance Sampling for Memory-Efficient Large Language Model Fine-Tuning. NeurIPS 2024.
> [2] ReFT: Representation Finetuning for Language Models. NeurIPS 2024.
> [3] MoRA: High-Rank Updating for Parameter-Efficient Fine-Tuning.
> [4] QuanTA: Efficient High-Rank Fine-Tuning of LLMs with Quantum-Informed Tensor Adaptation. NeurIPS 2024.
> [5] Mistral 7B. arXiv (2023).
> [6] Riemannian Preconditioned LoRA for Fine-Tuning Foundation Models. arXiv (2024).
> [7] Expressibility and entangling capability of parameterized quantum circuits for hybrid quantum‐classical algorithms. Advanced Quantum Technologies.

---

> ### Comment · Area_Chair_1MmG · 2024-11-27
>
> Dear Reviewer,
>
> The authors have provided their rebuttal to your comments/questions. Given that we are not far from the end of author-reviewer discussions, it will be very helpful if you can take a look at their rebuttal and provide any further comments. Even if you do not have further comments, please also confirm that you have read the rebuttal. Thanks!
>
> Best wishes,
> AC

---

> ### Comment · Reviewer_XPHL · 2024-11-27
>
> Thanks for the detailed rebuttal. The authors answered most of my questions. I'm still curious about the authors' choice of benchmarking on Mistral and DeBERTa but not LLaMA against the new references. Only [1] included benchmarks on Mistral but all [1] to [4] included benchmarks on LLaMA. None of the references included or made any claims about DeBERTa. It can be difficult to compare different works based on different benchmarks, and it is unclear from the provided benchmarks how Quantum-PEFT compares to these new methods on NLG tasks. Understandably, benchmarking fine-tuning methods is difficult and time-consuming. I'd be happy to raise the score if the authors could (qualitatively) explain how the benchmarks on Mistral and DeBERTa could translate to the benchmarks on LLaMA as shown in the references in order to better gauge the results, or if the authors could provide more theoretical reasoning on Quantum-PEFT's advantage over these methods.

---

> > ### Author Response · Authors · 2024-11-30
> >
> > We thank the reviewer for the reply.
> > We provide the following comparisons, empirical results on Llama, and further clarifications on our focus, as detailed below (will be included in the final version).
> >
> > **Comparisons.**
> > Quantum-PEFT's gains are rooted in fundamental mathematical principles, i.e., representing weight updates with highly compact unitary parameterizations, and therefore generally applicable independently of the underlying pre-trained model architecture.
> > This approach contrasts with LISA [1], which relies on layer-wise sampling motivated by LLM training behaviors.
> > LoReFT [2] leverages intervention in activation space at specific positions within the Transformer, which would require careful tuning per architecture and task.
> > MORA [3] highlights the importance of high-rank matrices in Llama benchmarks, where Quantum-PEFT uses underlying full-rank unitary parameterizations, which further suggests that Quantum-PEFT can also perform well in multiple tasks, including Llama benchmarks, while also benefitting from better-than-LoRA scaling unlike MORA's linear scaling.
> > QuanTA [4] does not consider unitary gains, while our SVD form offers enhanced regularization akin to AdaLoRA and improved training stability due to unitary normalization (e.g., [5]), both of which are beneficial when handling larger and deeper models with higher risk of overfitting and training instabilities, further suggesting favorable scaling of Quantum-PEFT to large models such as Llama applied to challenging NLG tasks requiring improved generalization capabilities.
> >
> > **NLG Llama evaluations.**
> > In response to the reviewer's feedback and to enable direct comparison with the provided references, we conducted additional experiments using Llama2-7B on arithmetic reasoning tasks by fine-tuning on MATH10K under the same settings as in QuanTA [4].
> > From the results, we observe that Quantum-PEFT achieves competitive performance with the newer methods on these math benchmarks while using fewer parameters.
> > These results on Llama benchmarks provide further empirical evidence for the broad applicability and efficiency of Quantum-PEFT.
> >
> >
> > | Method       | Params (%) | AQuA | GSM8K | MAWPS | SVAMP | Avg  |
> > | ------------ | -------- | ---- | ----- | ----- | ----- | ---- |
> > | LoRA         | 0.826    | 17.5 | 65.7  | 91.2  | 80.8  | 63.8 |
> > | LoReFT       | 0.031    | 18.0 | 60.8  | 89.1  | 76.2  | 61.0 |
> > | QuanTA       | 0.195    | 16.7 | 67.0  | 94.3  | 80.3  | 64.6 |
> > | MORA         | 0.824    | 17.2 | 64.5  | 90.8  | 80.5  | 63.3 |
> > | Quantum-PEFT | 0.017    | 16.9 | 66.9  | 93.9  | 80.9  | 64.7 |
> >
> >
> > **Focus**.
> > Our core contribution is proposing a new PEFT framework grounded in extremely efficient Pauli parameterization to enable logarithmic scaling of trainable parameters with unitary constraints, rather than mostly focusing on the specific application of LLM finetuning as done in [1,3,4].
> > The wide range of provided evaluations, including diverse language tasks on both well-established benchmarks and newer models, as well as vision tasks with vision transformers, demonstrate Quantum-PEFT effectiveness across multiple tasks and architectures.
> > The theoretical foundation, the demonstrated cross-domain efficacy, and the significant reduction in trainable parameters underscore the effectiveness of our contribution to the PEFT field.
> > We thank the reviewer for the feedback and we remain available for further discussions.
> >
> >
> > [1] LISA: Layerwise Importance Sampling for Memory-Efficient Large Language Model Fine-Tuning. NeurIPS 2024.
> > [2] ReFT: Representation Finetuning for Language Models. NeurIPS 2024.
> > [3] MoRA: High-Rank Updating for Parameter-Efficient Fine-Tuning.
> > [4] QuanTA: Efficient High-Rank Fine-Tuning of LLMs with Quantum-Informed Tensor Adaptation. NeurIPS 2024.
> > [5] Efficient Riemannian optimization on the Stiefel manifold via the Cayley transform. ICLR 2020.

---

> > > ### Comment · Reviewer_XPHL · 2024-11-30
> > >
> > > Thanks for providing the new results. I would like to raise my score based on the new empirical result.

---

### Official Review · Reviewer_nb9o · 2024-11-04

**Soundness:** 3
**Presentation:** 3
**Contribution:** 3
**Rating:** 6
**Confidence:** 3

**Summary:**

The paper introduces Quantum-PEFT, a quantum-inspired parameter-efficient fine-tuning (PEFT) framework for large language and vision models. Unlike traditional methods such as LoRA, which require low-rank adaptations of model weights, Quantum-PEFT uses quantum unitary parameterizations, resulting in a logarithmic scaling of parameters. Key contributions include the development of quantum-inspired modules using Pauli parametrization, significantly reducing the number of trainable parameters, while retaining competitive performance in various transfer learning benchmarks.

**Strengths:**

1. Quantum-PEFT’s application of quantum unitary parameterizations to PEFT is novel, differentiating it from conventional LoRA models.
2. It drastically reduces trainable parameters without performance loss and it is very important for the field, especially for resource-constrained training.

**Weaknesses:**

1. I'm not familiar with quantum ML, so it's a bit hard for me to understand the core concept. I think it's also hard to digest for someone who is not familiar with quantum ML. But this is fine because I don't think the intended audience of this paper includes someone not familiar with quantum ML.

**Questions:**

See above. It would be great if authors could add some self-contained introduction to the necessary quantum concepts used in the paper.

---

> ### Author Response · Authors · 2024-11-22
>
> We thank the reviewer for the valuable feedback and address all remaining concerns below.
>
> > Q1. It would be great if authors could add some self-contained introduction to the necessary quantum concepts used in the paper.
>
> While Quantum-PEFT draws inspiration from quantum computation principles, the proposed parameterization can also be purely understood within the context of classical linear algebra.
> In fact, quantum circuits represent unitary matrices as a product of small unitary matrices (i.e., quantum gates) with a total number of parameters growing logarithmically with the dimension.
> To further enhance clarity for readers less familiar with quantum computing concepts, in the updated manuscript (highlighted in blue), **we have revised the Introduction and the Preliminaries sections with more detailed and clearer self-contained explanation of the necessary background**. This includes an explanation of Pauli operators as matrices and the concept of quantum circuits as matrix transformations. We believe these explanations provide a more accessible entry point for a broader audience, including those unfamiliar with quantum machine learning. Specifically, the Preliminaries section includes a concise explanation of Pauli matrices, quantum gates (RY and CZ), and quantum circuits as transformations with unitary matrices.
> Furthermore, we have added Fig. 2 to the main text (previously in the appendix) to provide a visual aid for understanding the quantum-inspired PEFT modules. This figure illustrates the structure of the quantum circuit from its fundamental building gates, i.e., small unitary matrices. We have also improved the notations by improving the paragraph summarizing all notation used in the paper (beginning of Section 3) and added a table of symbols in Appendix A.7.

---

> > ### Comment · Reviewer_nb9o · 2024-11-30
> >
> > I thank authors for including more detailed preliminaries. After reading the revised version, I would like to keep my score as is and increase the confidence.

---

> ### Comment · Area_Chair_1MmG · 2024-11-27
>
> Dear Reviewer,
>
> The authors have provided their rebuttal to your comments/questions. Given that we are not far from the end of author-reviewer discussions, it will be very helpful if you can take a look at their rebuttal and provide any further comments. Even if you do not have further comments, please also confirm that you have read the rebuttal. Thanks!
>
> Best wishes,
> AC

---

### Author Response · Authors · 2024-11-25
**Summary of rebuttal as deadline is approaching**

Dear Reviewers,

Thank you for your constructive feedback. We highlight the main points of the rebuttal below, with detailed responses in the individual replies:

* **Accessibility for a broader audience**: We have revised the Introduction and Preliminaries sections to provide a more self-contained explanation of core concepts, including Pauli matrices, quantum gates (RY and CZ), and quantum circuits. We have also added Fig. 2 to main text to visually illustrate Eq. (2), and a table of symbols (Appendix A.7).
* **Evaluation on larger models**:  We have conducted new experiments on Mistral-7B. The results (Table XPHL-2) demonstrate that Quantum-PEFT both reduces trainable parameters (4.67x) and improves performance over LoRA and AdaLoRA.
* **Comparison with recent PEFT methods**: We have provided detailed comparative discussions and included additional experiments comparing with MORA and QuanTA (Table XPHL-3), demonstrating superior performance with significantly fewer parameters. We have also expanded the Related Work section discussing suggested recent works.
* **Choice of $L$**: We have included a sensitivity analysis of $L$ (Table XPHL-1) and added discussions on existing literature related to circuit depth and entanglement capacity in Appendix A.4. The results demonstrate a saturation effect beyond small $L$.
* **Learnable parameters**: We have clarified that in Eq. (2) the trainable parameters are the rotation angles $\theta_{k,l}$ associated with the RY gates, as also illustrated in Fig. 2.
* **Computational complexity**:
Overall complexity of our parameterizations remains highly competitive with LoRA. Empirical measurements show that Quantum-PEFT can substantially reduce the number of parameters at similar training times to baselines even with the mapping.
* **Gradient computation**: We demonstrated the feasibility of computing gradients in the Pauli parameterization and included a discussion on gradient computation, using PyTorch autograd, and convergence, referencing recent relevant literature in QML.


We believe the provided clarifications and additional experiments strengthen the paper's contributions.
We hope this clarifies the merits of our work.
As the discussion window is closing soon, we would be happy to address any further questions you may have.


Best regards,

The Authors

---

### Meta-Review · Area_Chair_1MmG · 2024-12-07

**Metareview:**

This paper proposed Quantum-PEFT, which leverages quantum computations for parameter-efficient fine-tuning (PEFT). The topic is interdisciplinary, and initially the reviewers raised various questions both from the perspective of machine learning and quantum computing, including accessibility for a broader audience, evaluation on larger models, comparison with recent PEFT methods, choice of L, computational complexity, etc. The authors did an excellent job during the rebuttal period, with various empirical experiments added to address these concerns. The scores from reviewers were subsequently increased to be all positive.

Therefore, the decision is to accept this paper at ICLR 2025. In any case, the authors should carefully merge all points raised during rebuttal to the final version of the paper.

**Additional Comments On Reviewer Discussion:**

During the rebuttal period, the authors made detailed responses for the questions asked by reviewers. In particular, the authors added plenty of new empirical results. This convinced Reviewer XPHL to raise the score to 6, and all scores have been positive at the end.

---

### Decision · Program_Chairs · 2025-01-22

Accept (Poster)